# Protein stabilization of ITF2 by NF-κB prevents colitis-associated cancer development

Mingyu Lee [1,2,3,4,12], Yi-Sook Kim [2,3,4,12], Suha Lim [2,3,4], Seung-Hyun Shin[5], Iljin Kim[6], Jiyoung Kim [2,4,7], Min Choi[2,3,4], Jung Ho Kim[8], Seong-Joon Koh[9], Jong-Wan Park [2,4,7] & Hyun-Woo Shin [2,3,4,7,10,11] ✉

Chronic colonic inflammation is a feature of cancer and is strongly associated with tumorigenesis, but its underlying molecular mechanisms remain poorly understood. Inflammatory conditions increased ITF2 and p65 expression both ex vivo and in vivo, and ITF2 and p65 showed positive correlations. p65 overexpression stabilized ITF2 protein levels by interfering with the binding of Parkin to ITF2. More specifically, the C-terminus of p65 binds to the N-terminus of ITF2 and inhibits ubiquitination, thereby promoting ITF2 stabilization. Parkin acts as a E3 ubiquitin ligase for ITF2 ubiquitination. Intestinal epithelial-specific deletion of ITF2 facilitated nuclear translocation of p65 and thus increased colitis-associated cancer tumorigenesis, which was mediated by Azoxymethane/Dextran sulfate sodium or dextran sulfate sodium. Upregulated ITF2 expression was lost in carcinoma tissues of colitis-associated cancer patients, whereas p65 expression much more increased in both dysplastic and carcinoma regions. Therefore, these findings indicate a critical role for ITF2 in the repression of colitis-associated cancer progression and ITF2 would be an attractive target against inflammatory diseases including colitis-associated cancer.

Although some cancers are due to germline mutations, the overwhelming majority are related to environmental factors, which have strong associations with chronic inflammation[1]. It has become evident that chronic inflammation is strongly correlated with tumorigenesis, and cancer has been described as "an injury that never restores[2]". Chronic inflammation is thought to be a major cause of morbidity and mortality globally because it substantially increases cancer risk[3,4].

Prolonged inflammation usually prompts a tissue injury-and-repair cycle, accelerates tissue remodeling, increases the propagation and clonal expansion of tumor-initiated cells, and can trigger mutagenic processes that serve as cancer-initiating events together with persistent oxidative damage inflicted by the inflamed microenvironment, thereby diminishing organ function[5]. Inflammation also affects cancer-related genes through epigenetic mechanisms that result in the

[1]Division of Allergy and Clinical Immunology, Brigham and Women's Hospital and Department of Medicine, Harvard Medical School, Boston, USA. [2]Department of Biomedical Sciences, Seoul National University Graduate School, Seoul, South Korea. [3]Obstructive Upper airway Research (OUaR) Laboratory, Department of Pharmacology, Seoul National University College of Medicine, Seoul, South Korea. [4]Cancer Research Institute, Seoul National University College of Medicine, Seoul, South Korea. [5]Hanmi Research Center, Hanmi Pharmaceutical Co. Ltd., 550 Dongtangiheung-ro Hwaseong-si 18469 Gyeonggi-do, South Korea. [6]Department of Pharmacology, Inha University College of Medicine, Incheon, South Korea. [7]Ischemic/Hypoxic Disease Institute, Seoul National University College of Medicine, Seoul, South Korea. [8]Department of Pathology, Seoul National University Hospital, Seoul National University College of Medicine, Seoul, South Korea. [9]Liver Research Institute and Seoul National University College of Medicine, Seoul, South Korea. [10]Department of Otorhinolaryngology-Head and Neck Surgery, Seoul National University Hospital, Seoul, South Korea. [11]Sensory Organ Research Institute, Seoul National University College of Medicine, Seoul, South Korea. [12]These authors contributed equally: Mingyu Lee, Yi-Sook Kim. ✉e-mail: charlie@snu.ac.kr

silencing of tumor suppressor genes[6]. Although chronic inflammation is a hallmark of cancer, its precise mechanism in cancer development remains unclear.

Colorectal cancer (CRC) has long been known as one of the best examples of a tight association between tumor and chronic inflammation, offering the possibility of characterizing novel methods for inflammation-induced cancer prevention. More specifically, the clearest link between inflammation and colon cancer has been found in patients with inflammatory bowel disease (IBD)-associated cancer[7]. CRC, the third leading cause of cancer-related death in the USA[8], most commonly occurs (sporadic CRC) by somatic mutations in genes encoding a component of the Wnt/β-catenin signaling pathway (e.g., APC, β-catenin, GSK-3β, or Axin), although it sometimes arises following sustained inflammation in the intestine, as in patients with IBD[9]. Although around only 5% of all CRC tumors progress in the context of obvious chronic inflammation, long-term exposure to inflammation in the intestine induces severe destruction of the intestinal mucosa and the formation of intractable ulcers, ultimately resulting in mortality in CRC patients[10]. Indeed, colitis-associated cancer (CAC) has a poorer survival rate than sporadic CRC in progressed and metastatic stages and accounts for 15% of mortality in IBD patients[11]. Nevertheless, the molecular mechanisms whereby inflammation contributes to cancer development remain obscure.

The major forms of idiopathic IBD comprise Crohn's disease (CD) and ulcerative colitis (UC), and severe inflammation increases the cumulative risk of colorectal cancer up to 2.9% and 18% by 10 years and 30 years respectively[12,13]. IBD is common, and its incidence is increasing as more regions of the world are becoming developed. Although the pathophysiology of IBD remains incompletely understood, dysregulated immune cells and cytokines have been implicated in persistent inflammation[13]. Among the risk factors associated with CAC development, inflammatory cytokines and related signaling pathways including tumor necrosis factor (TNF)/nuclear factor κB (NF-κB), interleukin (IL)−6/STAT3, COX2/PGE2, IL-23/Th17, IL-1β, and IL-22 pathways have been directly linked to the pathogenesis of CAC and can predispose CAC to progression[14]. One report showed that gain-of-function mutations in *TP53* are observed very early and can enhance TNF and NF-κB[15]. In addition, somatic mutations in genes such as *NFKBIZ*, *ZC3H12A*, *TRAF3IP2*, and *HNRNPF* have recently been identified in inflammatory conditions and considered as a risk factor for CAC development[16,17]. However, the exact molecular mechanisms responsible for inflammation-induced malignancy, to our knowledge, remain unclear.

Immunoglobulin transcription factor 2 (ITF2), which has also been referred to as E2-2, SEF-2, transcription factor 4 (TCF4), and bHLH19, belongs to the class I basic helix-loop-helix family, and primarily targets the E-box sequences in the promoters and enhancers of certain genes[18]. Although the function of ITF2 is mainly associated with neurodevelopmental disorders such as schizophrenia, intellectual disability, and Pitt-Hopkins syndrome, multiple studies have presented evidence for the complex role of ITF2 either as an oncogene or as a tumor suppressor in several types of cancer[19–21]. Many mucosal mRNA profiles derived from UC or CD patients have shown upregulated ITF2 expression, with positive correlations with TNF levels[22–24]. In particular, long-term exposure to cisplatin, which has been reported to induce NF-κB and TNF expression in various types of cancer cells, induces ITF2 deletion in non-small cell lung cancer (NSCLC), implying an association between ITF2 and inflammation[25]. Similar correlations have also been found in ovarian cancer and breast cancer[26,27]. The genes changed by ITF2 knockdown are related to apoptosis and the inflammasome[28]. Some reports have found ITF2 expression to be frequently suppressed in NSCLC, ovarian cancer, CRC, and other epithelial-origin tumors[20,29]. These studies prompted us to hypothesize that ITF2 might have a role in inflammation-induced cancers, such as CAC. However, the functional and mechanistic role of ITF2 in CAC development has not been investigated.

In the current study, we suggest that intestinal epithelial ITF2 is stabilized by p65 and plays a critical role in the regulation of NF-κB activation. Mechanistically, p65 binds to ITF2 by interfering with interactions with Parkin and suppresses ITF2 degradation by blocking ubiquitination, which in turn downregulates NF-κB downstream genes and inhibits CAC progression. This study uncovers the mechanism underlying the role of ITF2 in the progression of inflammation-induced cancers such as CAC.

## Results

### ITF2 expression is dependent on p65

To determine whether ITF2 expression differed in inflammatory conditions in epithelial cells, we employed an experimental murine acute colitis model. Intestinal injury promoted by DSS upregulated the ITF2 expression in the colonic epithelial cells, as confirmed by immunoblotting (Fig. 1A). We then applied an intestinal organoid culture (known as mini-guts or enteroids) to directly explore whether ITF2 induction occurred in the intestinal epithelial cells by major inflammatory cytokines such as TNF without potential interference from the tissue microenvironment. We observed significant growth of the colonic crypt-derived organoids over 10 days of ex vivo culture (Fig. 1B and Supplementary Fig. S1A, B). Similarly, ITF2 expression was reproduced ex vivo in a medium including Wnt-3a, EGF, noggin, and R-spondin (WENR medium) for 3D mini-gut organoid culture followed by TNF treatment (Fig. 1B). To understand the precise molecular mechanism by which TNF induces ITF2 expression, we initially selected appropriate cell lines. ITF2 was found to be hardly expressed in most human colon cancer cell lines, except for Colo320DM, SW480, CaCo2, and WiDr (Supplementary Fig. S2A). In ITF2-deficient cell lines, including DLD-1 and HT-29, ITF2 restoration with TNF treatment increased ITF2 expression in accordance with NF-κB activation (Supplementary Fig. S2B). TNF delivery consistently upregulated ITF2 expression in ITF2-expressing cell lines such as Colo320DM and CaCo2 (Supplementary Fig. S2C). We then investigated which signaling pathways are associated with the TNF-mediated ITF2 expression by using representative known inhibitors such as mTOR (rapamycin), MEK/ERK (PD98059), AKT/p38 (SB203580), JAK-STAT (AG490), AKT (MK2206) and NF-κB (Bay11-7082). The NF-κB inhibitor (Bay11-7082) significantly blocked TNF-induced ITF2 expression (Fig. 1C). For confirmation, we additionally added inhibitors (Bay11-7085, TPCA1) to block NF-κB and then examined whether this finding is repeatedly observed in different colon cancer cell lines. The SW480, WiDr, and HCT-15 cell lines including Colo320DM which can express ITF2 showed a similar result (Fig. 1D and Supplementary Fig. S3A, B). Although NF-κB inhibitors did not result in a noteworthy decrease in p65, they exhibited a reduction in a pattern following p65 inhibition, as shown in Fig. 1C, D, and Supplementary Fig. S3. Thus, the question arose of whether p65 itself induces ITF2 expression. To answer this question, we overexpressed ITF2 with p65 in a concentration-dependent manner and found robust induction of ITF2. However, ectopic expression of p50 did not affect ITF2 induction (Fig. 1E). We further overexpressed a p65 plasmid in colon cancer cell lines and noticed that p65 induced the expression of endogenous ITF2 (Fig. 1F). To confirm these observations in vivo, we examined the relationship between p65 and ITF2 in tissues from DSS-treated mice by utilizing immunohistochemistry. We confirmed representative colitis markers such as colon length, body weight, and morphology of the colonic tissues (Supplementary Fig. S4A–D). In the tissues, the levels of ITF2 and p65 were significantly upregulated by DSS treatment (Fig. 1G–I). Moreover, immunohistochemistry results revealed a marked positive correlation between ITF2 and p65 (Fig. 1J). TNF-induced ITF2 expression was blocked in stable ITF2-knockdown cell lines (Fig. 1K). Collectively, these results support the hypothesis that p65, not p50, is required for ITF2 expression.

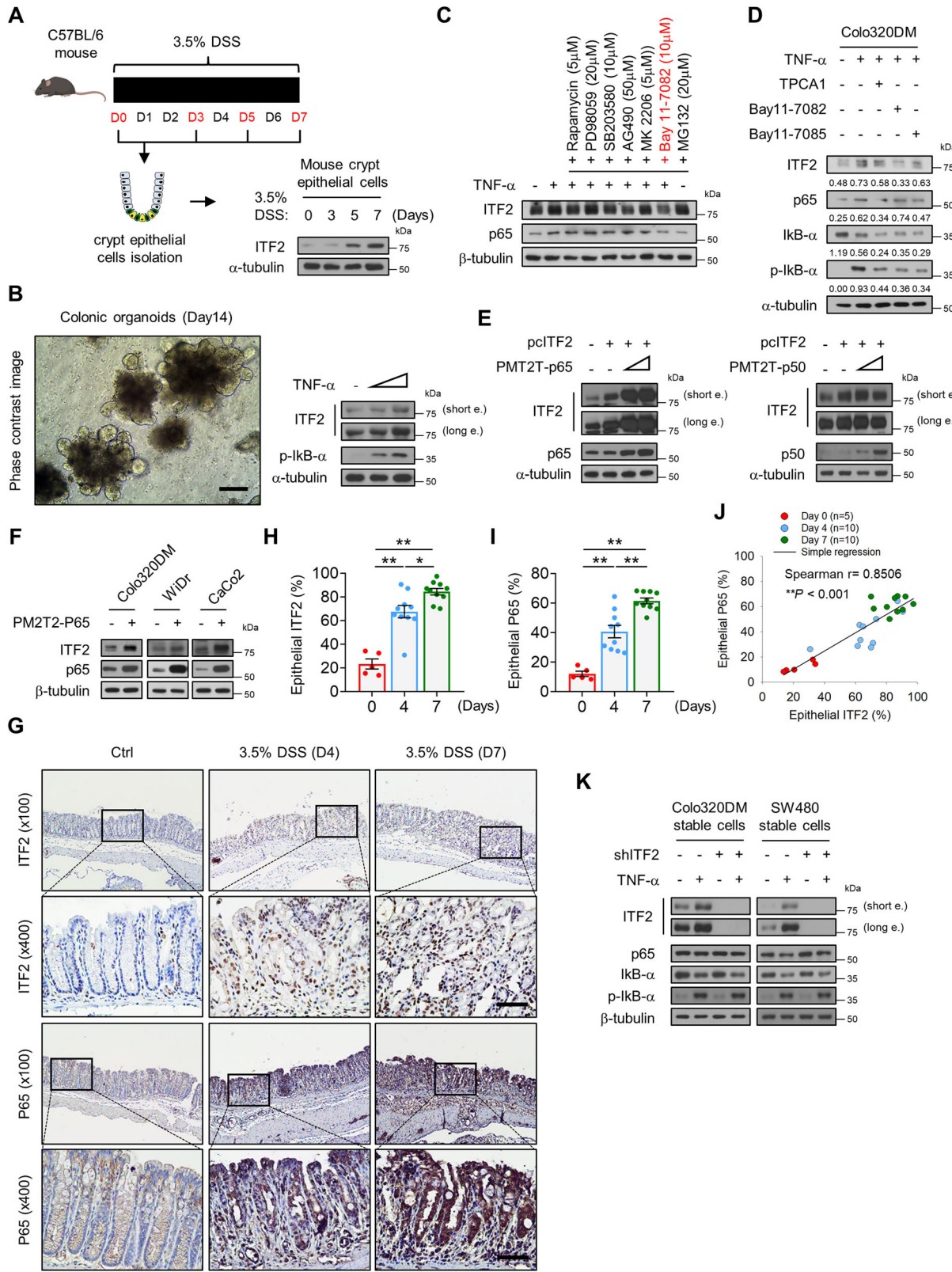

## p65 promotes ITF2 stabilization by inhibiting its ubiquitination

To understand how p65 regulates ITF2 expression, we first inspected the mRNA levels of ITF2 in TNF-treated colon cancer cells in a time-and concentration-dependent manner. ITF2 mRNA levels did not change in the presence of TNF, even at high concentrations (Fig. 2A). In addition, ITF2 mRNA expression was comparable following TNF treatment in both control and siRNA-induced p65-depleted cells (Fig. 2B). However,

p65 knockdown reduced endogenous ITF2 protein levels (Fig. 2C). Ectopic expression of p65 consistently showed no impact on ITF2 mRNA levels (Fig. 2D). We then explored whether TNF affects the protein stability of ITF2 by evaluating the protein half-life of ITF2. Colo320DM cells were stimulated with either vehicle or TNF for 8 h before they were treated with CHX, a protein synthesis inhibitor, for different periods. Cells treated with TNF exhibited a higher half-life of

**Fig. 1 | ITF2 expression is attributable to p65. A** Schematic diagram of DSS treatment to induce acute colitis in the mouse intestine. Western blot analysis of ITF2 in the colonic epithelial cells of WT mice treated with 3.5% DSS for 0–7 days. Mouse icon was created with BioRender (BioRender.com). $n = 2$, biologically independent experiment. **B** Phase contrast image (left panel) and western blot analysis of ITF2 in the colonic organoids treated with TNF (20 ng/ml) in a concentration-dependent manner (right panel, $n = 2$, biologically independent experiment). **C** Colo320DM cells were pretreated with the indicated inhibitors for 1 h, followed by TNF stimulation for 8 h. MG132 was treated without TNF. Specified protein expressions were traced. $n = 3$, biologically independent experiment. **D** Colo320DM cells were pre-incubated with indicated NF-κB inhibitors for 1 h and cultured with TNF for 8 h. The indicated proteins were traced and presented as immunoblots. Each data are expressed as normalized band intensity adjusted to α-tubulin, which serves as a loading control. The protein intensities were quantified by ImageJ software. $n = 3$, biologically independent experiment. **E** Colo320DM cells were overexpressed with p65 and p50, representative NF-κB subunits, in a concentration-dependent manner with or without ITF, and then ITF2 expression

was evaluated. $n = 2$, biologically independent experiment. **F** Colon cancer cells were transfected with p65 and ITF2 expression was compared. $n = 3$, biologically independent experiment. **G** Representative immuno-stained images of ITF2 and p65 expression in acute colitis-induced colon tissues. The data are representative of three independent experiments. **H** Epithelial expression levels of ITF2 were analyzed and compared ($n = 5, 10, 10$ respectively, biologically independent animals., **$P = 0.0027$, **$P = 0.0007$, *$P = 0.0122$). **I** Epithelial expression levels of p65 were analyzed and compared ($n = 5, 10, 10$ respectively, biologically independent animals., **$P = 0.0007$, **$P = 0.0006$, **$P = 0.0006$). **J** Linear regression between ITF2 and p65 in each group of mice (Spearman $r = 0.8506$, **$P < 0.001$). **K** Stable cell lines were established with small hairpin RNA targeting ITF2 and treated with or without TNF. The indicated proteins were traced and presented as immunoblots. $n = 2$, biologically independent experiment. Scale bars, 100 µm. In all immunoblot analyses, the data are representative of three independent experiments. All results are presented as means ± SEM. Statistical significance was determined by the two-tailed Mann–Whitney $U$ test (**H**, **I**) and the two-tailed Spearman correlation test (**J**). Source data are provided in the Source Data file.

---

ITF2 protein than vehicle-treated cells (Fig. 2E). Similar findings were observed in cells co-transfected with Myc-ITF2B and p65 (Fig. 2F). As TNF and CHX incubation could induce RIP1-independent apoptosis in the cells, we simply examined whether ITF2 affects TNF-induced cell death pathways. Annexin V-PI staining results presented that there are any significant changes across the samples in the early time point, but the proportion of apoptotic cells was highly reduced in 24 h stimulatory condition. (Supplementary Fig. S5). We observed that a large amount of the ITF2 protein was regulated through the ubiquitin-proteasome degradation system by treating four colon cancer lines with the proteasome inhibitor MG132 (Fig. 2G). To verify whether p65 promotes ITF2 stabilization through ubiquitination, in vivo ubiquitination assays were performed. HEK293T cells were co-transfected with a control vector or Flag-p65 together with vectors expressing Myc-ITF2B and HA-Ubiquitin (HA-Ub). Cells with Flag-p65 expression displayed lower ubiquitination of Myc-ITF2 than cells transfected with the control vector (Fig. 2H). An analysis of several public microarray datasets supported our finding that ITF2 mRNA levels did not differ between normal samples and those from UC or CD patients (Fig. 2I). β-catenin has been reported to activate various downstream genes, including ITF2[30]. Therefore, we tested whether ITF2 stabilization could be attributed to β-catenin activation. Although colon cancer cells that were co-transfected with TOP-flash and FOP-flash reporter plasmids were either stimulated with TNF or overexpressed with the p65 plasmid, we did not detect any significant changes in terms of β-catenin activity (Supplementary Fig. S6A, B). In addition, ITF2 expression was further augmented by TNF even after β-catenin knockdown (Supplementary Fig. S6C, D). Taken together, our results indicate that TNF stimulation or p65 overexpression reduced ITF2 ubiquitination regardless of β-catenin activation, suggesting that ITF2 is stabilized at the protein level.

### p65 binds ITF2 and attenuates ITF2 ubiquitination

Protein stabilization can occur through protein-protein binding by masking the ubiquitination site. We, therefore, carried out a co-immunoprecipitation experiment to check for direct interactions between ITF2 and p65. ITF2 is co-immunoprecipitated by the p65 antibody, but not by control IgG (Fig. 3A). An in vitro binding assay showed similar observations (Fig. 3B). To investigate which domain of p65 is responsible for ITF2 ubiquitination, we generated plasmids that could express the domains of each protein and conducted immuno-precipitation (Supplementary Fig. S7A–C). p65 was found to bind to the N-terminal fragment of ITF2B (Fig. 3C). We also found that ITF2 interacted with the C-terminus of p65 (Fig. 3D). In vitro binding analyses further supported binding between the N-terminus of ITF2 and the C-terminus of p65 (Fig. 3E and Supplementary Fig. S7C). Since the middle part of β-catenin also can bind to the N-terminus of ITF2[20], we

hypothesized that p65 and β-catenin compete with each other for binding to ITF2. p65 interacted with ITF2 but reduced the binding of β-catenin to ITF2. Conversely, β-catenin overexpression dissociated p65 from ITF2 (Fig. 3F). These data imply that ITF2 prefers p65, which is overexpressed in dysplasia and CAC tissues, as a binding partner instead of β-catenin. We further scrutinized whether the C-terminal of p65 is necessary for the downregulation of ubiquitination of the N-terminus of ITF2. As expected, we observed that only the C-terminus of p65, not the N-terminus, reduced the ubiquitination of the N-terminus of ITF2 (Fig. 3G). Ectopic expression of p65 consistently increased expression of the N-terminus of ITF2 (Fig. 3H). As RelB and c-Rel reportedly, not p50 and p52, have similar domain structures to that of p65 among five NF-κB factors, we furtherly evaluated whether RelB and c-Rel are associated with the stabilization and ubiquitination of ITF2 as p65 did. Even though RelB or c-Rel bind to the ITF2, they cannot affect the expression and ubiquitination levels of the ITF2 (Supplementary Fig. S8A–C). These results suggest that the C-terminus of p65, not its N-terminus, is necessary for interactions with the N-terminus of ITF2, resulting in the downregulation of ITF2 ubiquitination.

### Parkin targets ITF2 for ubiquitination

Although ITF2 can be degraded through the ubiquitin-proteasomal degradation system according to the previous results (Fig. 2G), an E3 ubiquitin ligase, which is responsible for ITF2 ubiquitination, is still unidentified. To identify potential candidates engaged in ITF2 ubiquitination, we surveyed a protein-protein interaction database. Among the 284 unique ITF2-interacting proteins displayed in the BioGRID, we found 9 candidates including Parkin which have an E3 ubiquitin-protein ligase role (Fig. 4A), and then we narrowed down to the 3 promising targets based on the experimental evidence which was demonstrated by the two-hybrid system. However, the knockdown experiment presented that only Parkin was responsible for the ITF2 ubiquitination (Supplementary Fig. S9A–C). Reversely, overexpression of Parkin with ITF2 in the HeLa cells which have little or no endogenous Parkin expression[31,32], induced ITF2 ubiquitination (Supplementary Fig. S9D). Co-immunoprecipitation and in vitro binding assay confirmed that Parkin can bind to ITF2 (Fig. 4B, C), and it turns out to be that Parkin binds to the N-terminals of ITF2B (Fig. 4D). To corroborate which sites of ITF2 are associated with Parkin-induced ITF2 ubiquitination, we generated an ITF2 mutant where key lysine residues (Lys171-175) are removed based on the AlphaFold2 results (Fig. 4E). As expected, when the HeLa cells are overexpressed with Parkin and WT ITF2, we observed upregulation of ITF2 ubiquitination. However, transfection with ITF2 mutant and Parkin revealed attenuated ubiquitination capacity. More specifically, the K48-linked ubiquitin chain is engaged in the Parkin-mediated ITF2 ubiquitination (Fig. 4F). The

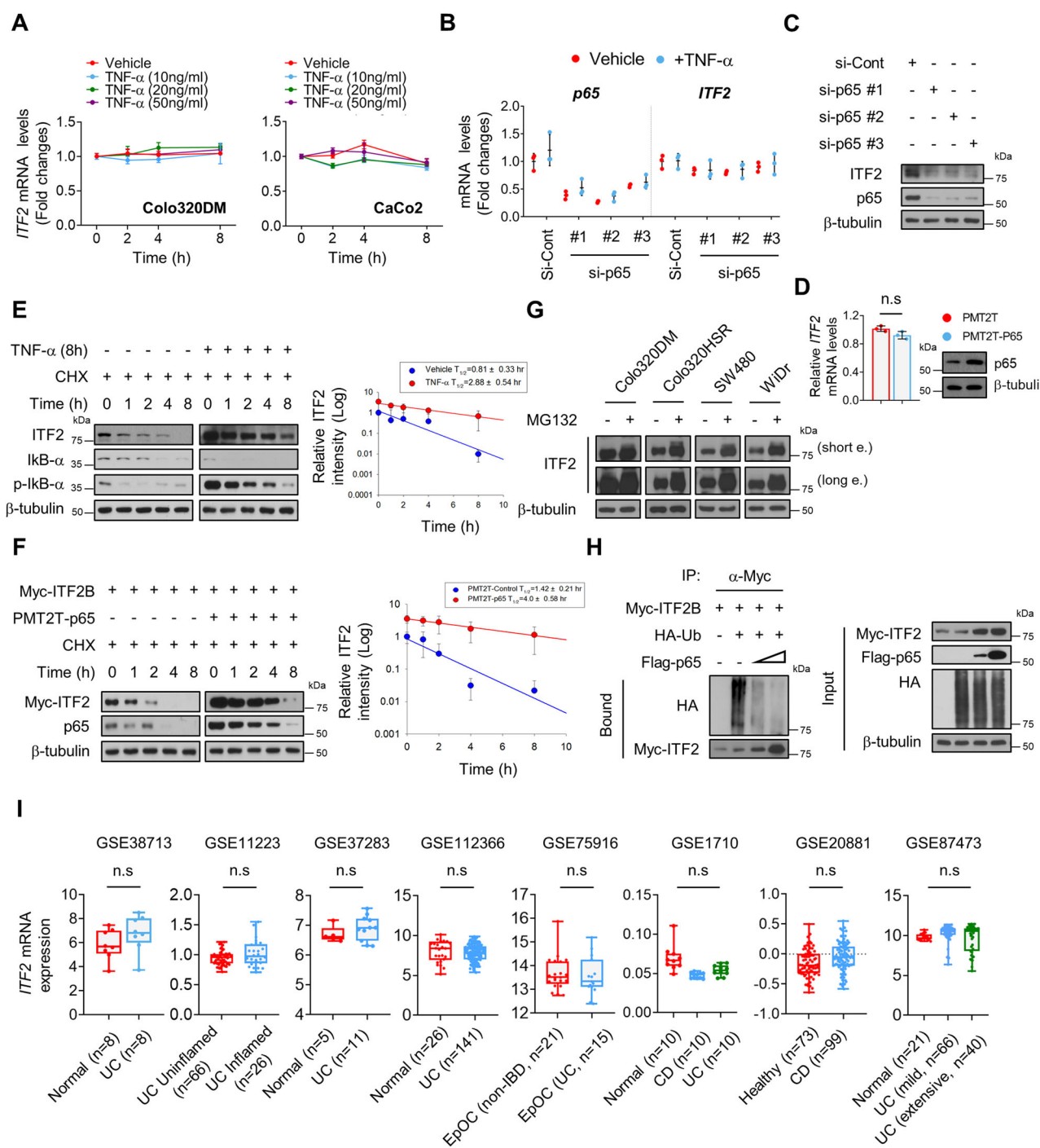

ubiquitination of ITF2 by Parkin was examined by in vitro ubiquitination assays using purified Myc-Parkin and F/S-ITF2 proteins. Myc-Parkin promoted the ubiquitination of F/S-ITF2B in the presence of recombinant E1, E2, ubiquitin, and ATP in vitro (Supplementary Fig. S9E). Considering that p65 decreases ITF2 ubiquitination by interacting with ITF2, we questioned whether p65 binding inhibits interactions between Parkin and ITF2. Interestingly, p65 competed with Parkin for the binding site of ITF2 (Fig. 4G). Finally, we investigated whether ITF2 ubiquitination is decreased in the presence of the C-terminus of p65 or Parkin knockdown conditions. While the N-terminus of p65 overexpression did not affect ITF2 ubiquitination, full-length p65 or the C-terminus of p65 is required for ITF2 ubiquitination. However, siRNA knockdown targeting Parkin abolished the ubiquitination of ITF2 in cells transfected with the N-terminus of p65

(Fig. 4H). Consistently, co-immunoprecipitation assay against endogenous proteins as well as PLA assay indicated that ITF2 interacts with p65 and Parkin (Supplementary Fig. S10A–C). The endogenous interaction between p65 and ITF2 was notably increased following the TNF stimulation (Supplementary Fig. S10D). In addition, the levels of immunoprecipitated p-p65, a hallmark of NF-κB activation, by the ITF2 are augmented after TNF treatment (Supplementary Fig. S10E). Taken together, these results reveal that Parkin competes with p65 for the binding of ITF2, resulting in ubiquitination.

**Anti-tumorigenic effects of ITF2 in inflammatory condition**
Because p65 is one of the components of NF-κB and is highly associated with inflammation and inflammation-induced diseases, we then explore whether the stabilization of ITF2 is required for colon cancer

**Fig. 2 | p65 stabilizes ITF2 expression at the protein level. A** Colo320DM and CaCo2 cells were treated with TNF in a time-and concentration-dependent manner, and ITF2 mRNA levels were traced (*n* = 3, biologically independent experiments). **B** Real-time qRT-PCR to analyze the expression of ITF2 mRNA. Colo320DM cells were transfected with si-RNA targeting p65 (60 nM) and then treated with or without TNF (20 ng/ml) for 8 h. Scrambled siRNA sequence was used as a control (*n* = 3, biologically independent experiments). **C** Colo320DM cells were transfected with si-Cont (60 nM) or si-p65 (60 nM), followed by TNF treatment for 8 h. ITF2 expression levels were compared. *n* = 2, biologically independent experiment. **D** Cells were transfected with a p65 plasmid and lysed for ITF2 mRNA comparison. *n* = 2, biologically independent experiment. **E** TNF treatment increased the ITF2 protein half-life in Colo320DM cells. Colo320DM cells were pre-treated with either vehicle or TNF for 8 h, and incubated with 100 μM CHX for the indicated periods before being collected for western blot analysis. *n* = 3, biologically independent experiment. **F** p65 overexpression maintained the Myc-ITF2 protein half-life in cells. Colo320DM cells with ectopic p65 expression and control cells were transfected with a Myc-ITF2 plasmid vector. The cells were treated with CHX (100 μM) for the indicated time points. *n* = 3, biologically independent experiment. **E**, **F** The ITF2

protein levels were analyzed by western blot analysis with β-tubulin as a loading control (left). Protein levels were measured with the densitometric intensity. ITF2 levels were quantified relative to β-tubulin levels, and the half-life was calculated (right). **G** Colon cancer cells were treated with MG-132 (20 μM) for 8 h, and endogenous ITF2 expression was examined by immunoblotting. *n* = 2, biologically independent experiment. **H** An experiment examined the effects of the expression of Flag-p65 on the ubiquitination of Myc-ITF2 and auto-ubiquitination of ITF2 in HEK293T cells. *n* = 3, biologically independent experiment. **I** ITF2 mRNA expression levels were compared between healthy samples and patients with UC or CD in various datasets. In boxplot (**I**), the central horizontal line shows the median; the upper and lower bounds of box show third and first quartile; the upper and lower horizontal lines show the maxima and minima of the data. In all immunoblot analyses, the data are representative of three independent experiments. All results are presented as means ± SEM. Statistical significance was determined by the unpaired two-tailed Student *t* test (**A**, **B**, **D**) and the Kruskal–Wallis test, followed by the two-tailed Mann–Whitney *U* test (**I**) for pairwise comparisons. NS not significant. Source data are provided in the Source Data file.

development mediated by inflammation. We first generated stable cell lines transfected with shRNA targeting ITF2 from the Colo320DM and SW480 cell lines (ITF2-expressed cell lines) and established stable FL-ITF2 overexpressed cells from the DLD-1 and HCT116 cell lines (ITF2-null cell lines). We then assessed the levels of key pro-tumorigenic cytokines related to CAC development. Upregulation of pro-tumorigenic genes following ITF2 suppression was confirmed by a quantitative reverse-transcription polymerase chain reaction in stable ITF2-knockdown cells versus control. Conversely, the transcripts of pro-tumorigenic cytokines were downregulated by ITF2 restoration in stable ITF2-overexpressed cells (Fig. 5A, B). Since most pro-tumorigenic cytokines are downstream genes of NF-κB, we speculated that ITF2 might act as an NF-κB suppressor by decreasing p65 translocation to the nucleus. Based on this hypothesis, we stably transfected HCT116 cells with empty vector, full-length ITF2 or N-terminus ITF2, and performed a nuclear fractionation experiment. Interestingly, the levels of p65, which was translocated into the nucleus, were downregulated in response to full-length ITF2 or N-terminus ITF2 overexpression, as shown by immunoblotting (Fig. 5C). We also conducted a similar experiment under TNF-treated conditions to mimic inflammatory conditions. We observed that plenty of p65 was left in the cytosol even after TNF treatment, implying that p65 has a chance for physiological binding with ITF2 in the cytosol under inflammatory conditions. In addition, the amount of translocated p65 to the nucleus by the TNF challenge was markedly downregulated by the ITF2 overexpression (Fig. 5D). These results suggest that ITF2 could function as a suppressor of inflammation-mediated cancer development.

### ITF2 loss promotes AOM/DSS-induced tumor development
Next, we further extended our findings in vivo. To address whether epithelial ITF2 loss could exacerbate colonic tumorigenesis, we generated mice with ITF2 deficiency only in colonic epithelial cells (termed ITF2^ΔIE) by crossing *ITF2*-floxed mice with Villin-cre mice (Fig. 6A), confirmed by western blot, H&E staining, and immunohistochemistry (Fig. 6B, C). Although ITF2 loss could not affect any morphological changes and colon length in endogenous conditions (Supplementary Fig. S11A–C), ITF2^ΔIE mice were more susceptible to chronic inflammation, which was induced by AOM and DSS treatment, as confirmed by the loss of body weight, survival ratio, colon length, tumor numbers, and tumor sizes (Fig. 6D–J). The polyp size distribution data denoted that ITF2^ΔIE mice treated with AOM/DSS exhibited larger-sized tumors compared to littermates (Fig. 6K). Based on the information that ITF2 interacts with p65 and inhibits NF-κB activation in vitro (Fig. 5C), we counted the nuclear-positive p65 numbers in AOM/DSS-induced colonic tumor tissues from both ITF2^ΔIE mice and littermates.

Nuclear translocation of p65 consistently appeared to be increased in tissues from ITF2^ΔIE mice, as demonstrated by immunohistochemistry against ITF2 and p65 antibodies (Fig. 6L, M). Consistently, the NF-κB target genes such as IL-6 and TNF were highly increased in both non-tumor and tumor areas of the ITF2^ΔIE mice in contrast with the littermates. However, Parkin and cleaved caspase-3 (one of the apoptosis markers) expression levels were comparable across the samples (Supplementary Fig. S12A, B). As previously reported, AOM injection induces mutations (e.g., in *TP53*) and affects β-catenin activity[33], we, therefore, performed a similar in vivo experiment without AOM injection (Supplementary Fig. S13A). Surprisingly, dysplasia or microadenoma-like regions were frequently detected in ITF2^ΔIE mice with decreased colon length (Supplementary Fig. S13B–E). Similarly, the number of p65 nuclear-positive cells was elevated in the ITF2^ΔIE mice (Supplementary Fig. S13F, G). Taken together, these observations demonstrate that ITF2 loss increases susceptibility to inflammation in colonic epithelial cells, contributing to inflammation-mediated tumor development.

### Reciprocal expression of ITF2 and p65 in CAC tissues
To obtain a better understanding of the relationships among ITF2, β-catenin, and p65 during CAC progression, 22 human CAC samples, each of which contained normal, dysplastic, and carcinoma areas on the same slide, were assessed by immunohistochemistry. When we compared the expression of key molecules between CAC and sporadic CRC patients, in the sporadic CRC patient tissues, β-catenin activity was already upregulated in the adenoma stages and was further elevated in the carcinoma regions, suggesting that β-catenin is the main driver of sporadic CRC progression, as reported previously (Supplementary Fig. S14A, B). However, nuclear β-catenin was hardly detected in the normal and dysplastic regions of the CAC tissues, implying that other molecules other than β-catenin could drive inflammation-induced colon cancer development (Fig. 7A, B). In contrast, p65 expression was increased in the dysplastic tissues and was substantially upregulated in the carcinoma areas. Interestingly, ITF2 expression increased in dysplastic tissues but was finally lost in carcinoma regions, as detected in sporadic CRC patients (Fig. 7A, B and Supplementary Fig. S14A, B). We, thus, questioned whether ITF2 is associated with the disease stage in patients with UC and CAC. After obtaining additional 15 UC specimens, we stained ITF2 and then compared the disease stage based on ITF2 expression levels. The patients in the late stage (stages 3 and 4) presented significant loss of ITF2 (Supplementary Fig. S15A, B), suggesting that ITF2 is highly correlated with CAC disease stage. However, Parkin levels are comparable between dysplastic and carcinoma regions of the CAC patient tissues (Supplementary Fig. S15C). Consistent with this finding, the GSE3629

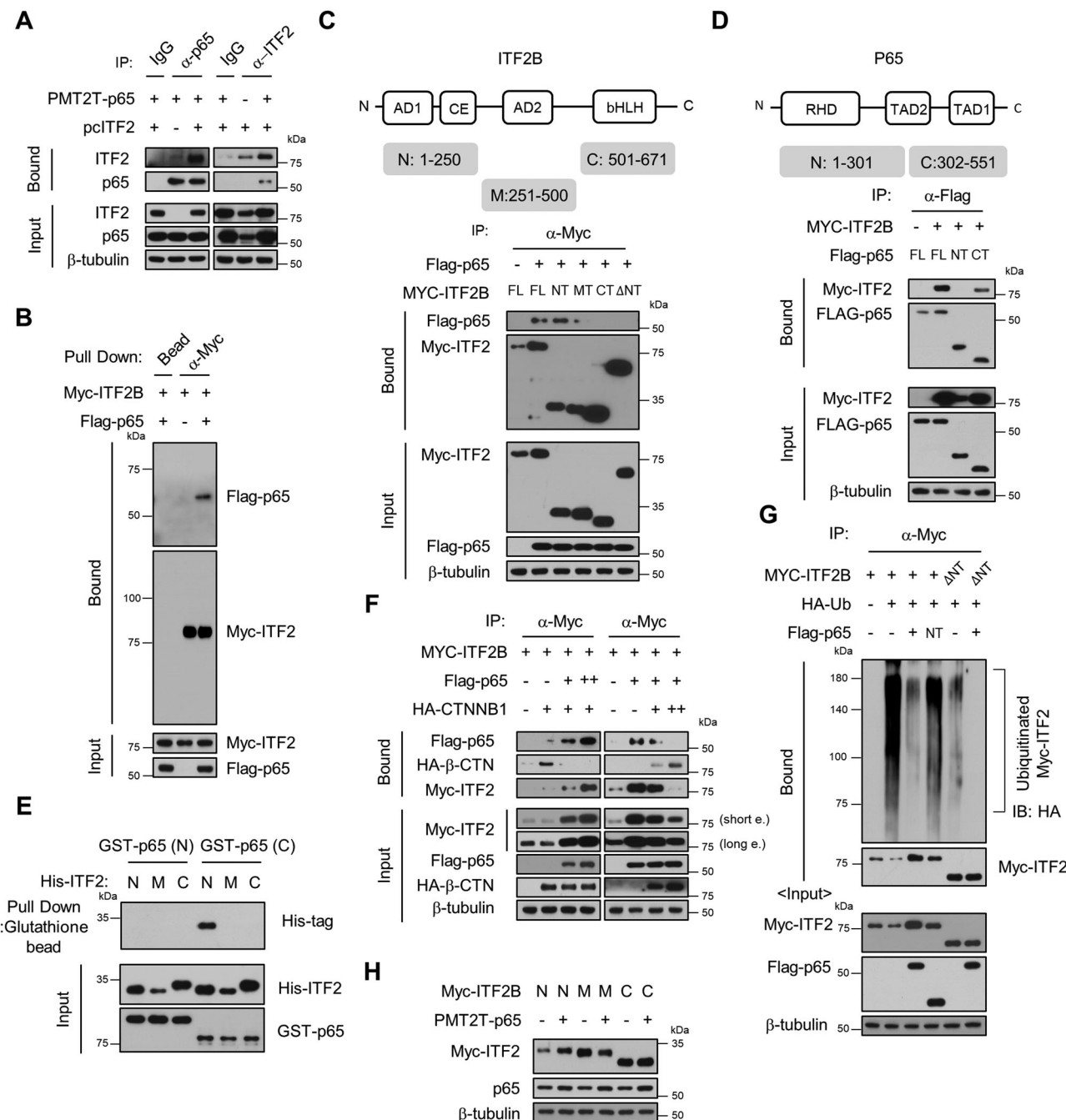

**Fig. 3 | The C-terminal of p65 binds the N-terminal of ITF2 and downregulates ITF2 ubiquitination. A** ITF2 interacts with p65. Cells were transfected with either p65 or ITF2 vector or both. Protein interactions were analyzed by immunoprecipitation-immunoblotting. $n = 2$, biologically independent experiment. **B** In vitro binding analysis. Purified Myc-ITF2 and Flag-p65 proteins were mixed in a test tube, and pulled-down proteins were immunoblotted. $n = 3$, biologically independent experiment. **C** The N-terminal of ITF2 interacts with p65. Each Myc-tagged ITF2B fragment was co-expressed with Flag-p65 in HEK293T cells. After being immuno-precipitated with α-Myc, proteins were analyzed using α-Flag. $n = 2$, biologically independent experiment. **D** The C-terminal of p65 is necessary for it to bind to ITF2. Each Flag-tagged p65 fragment was co-expressed with Myc-ITF2B in HEK293T cells. After being immuno-precipitated with α-Flag, proteins were examined using α-Myc. $n = 2$, biologically independent experiment. **E** In vitro binding analysis. Recombinant His-ITF2 and GST-P65 peptides, which had been

isolated from *E. coli*, were placed together in a test tube and mixed in $2 \times 3$ combinations. His-ITF2 was pulled down using glutathione-affinity beads and subjected to immunoblotting. $n = 3$, biologically independent experiment. **F** Competitive interaction between P65 and β-catenin for binding ITF2B. Myc-ITF2B was co-transfected into HEK293T cells with HA-β-catenin and increasing (1, 3 μg) Flag-p65 amounts (left). Myc-ITF2B (0.5 μg), Flag-p65 (0.5 μg), and HA-β-catenin (1, 3 μg) were co-transfected (right panel). The protein interactions were analyzed by immunoprecipitation-immunoblotting. $n = 2$, biologically independent experiment. **G** The effects of the expression of Flag-p65 (FL, NT) on ubiquitination of Myc-ITF2B (FL, ΔNT) in HEK293T cells were analyzed by in vivo ubiquitination assays. $n = 3$, biologically independent experiment. **H** The indicated Myc-ITF2B domains were co-transfected with p65 and analyzed by western blot. $n = 2$, biologically independent experiment. In all immunoblot analyses, the data are representative of three independent experiments. Source data are provided in the Source Data file.

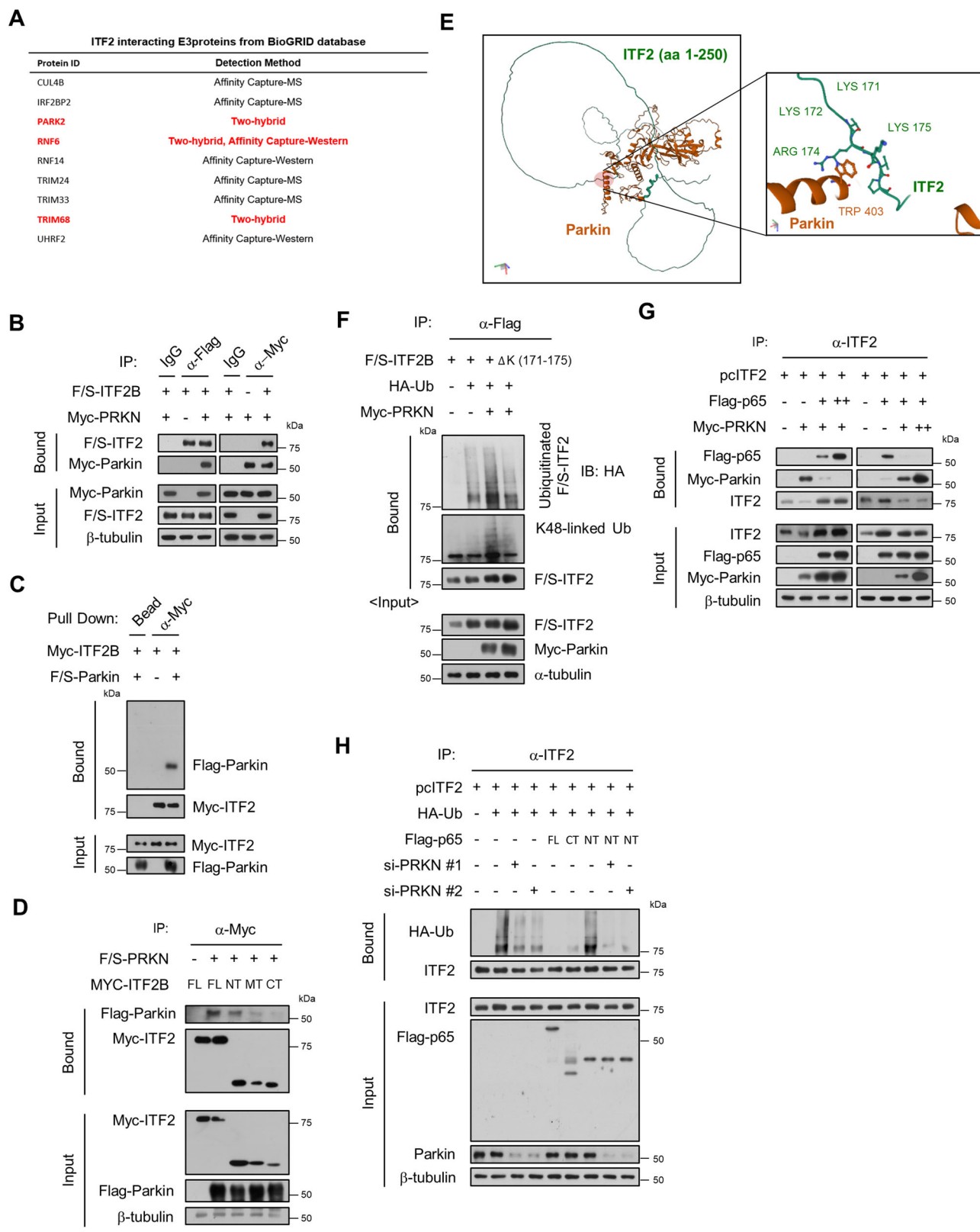

dataset showed scarce expression of ITF2 in the UC-associated neoplastic legions, sporadic CRC tissues, and UC-associated carcinoma tissues (Fig. 7C). To summarize, NF-κB activity is already upregulated at the adenoma stage and is activated further along with ITF2 suppression at the carcinoma stage, which further supports our hypothesis that ITF2 loss renders tumors more malignant by increasing NF-κB activation.

## Discussion

The incidence of colitis-associated cancer (CAC) in patients with IBD (e.g., UC and CD) continues to be higher than in the general population, and CAC shows a poorer survival rate, with higher NF-κB expression, than sporadic CRC. The course and underlying genetic events of IBD-associated neoplasia differ markedly from those of sporadic CRC. Sporadic CRC is characterized by adenomatous polyps,

**Fig. 4 | Parkin promotes ubiquitination and degradation of ITF2. A** A list of the ITF2-interacting E3 proteins from the BioGRID database. **B** ITF2 interacts with Parkin. HEK293T cells were transfected with either F/S-ITF2B or Myc-PRKN vector or both. Protein interactions were analyzed by immunoprecipitation-immunoblotting. *n* = 3, biologically independent experiment. **C** In vitro binding analysis. Purified Myc-ITF2B and F/S-Parkin proteins were mixed in a test tube, and pulled-down proteins were immunoblotted. *n* = 2, biologically independent experiment. **D** The N-terminal of ITF2 interacts with Parkin. Each Myc-tagged ITF2B fragment was co-expressed with F/S-PRKN in HEK293T cells. After being immuno-precipitated with α-Myc, proteins were analyzed using α-Flag. *n* = 2, biologically independent experiment. **E** Parkin binds to the N-terminal of ITF2 between Lys residues (171–175) based on the AlphaFold2 (AF2) prediction result. **F** Plasmids encoding WT F/S-ITF2B or F/S-ITF2B-ΔK (171–175) mutant were co-transfected with

HA-Ub into HEK293T cells in the absence or presence of Myc-PRKN. Cell lysates were immunoprecipitated with α-Flag antibodies and immunoblotted. *n* = 3, biologically independent experiment. **G** Competitive interaction between p65 and Parkin for binding with ITF2B. pcITF2 was co-transfected into HEK293T cells with Myc-PRKN and increasing (1, 3 µg) Flag-p65 amounts (left). pcITF2 (0.5 µg), Flag-p65 (0.5 µg) and Myc-PRKN (1, 3 µg) were co-transfected (right panel). The protein interactions were analyzed by immunoprecipitation-immunoblotting. *n* = 3, biologically independent experiment. **H** The effects of Flag-p65 overexpression and PRKN knockdown on ITF2 ubiquitination. Each Flag-tagged p65 fragment was co-expressed with pcITF2 in HEK293T cells with or without siRNA targeting PRKN. *n* = 3, biologically independent experiment. In all immunoblot analyses, the data are representative of three independent experiments. Source data are provided in the Source Data file.

a small percentage of which progress to carcinoma, whereas IBD-associated CRC often includes flat dysplastic lesions that develop from low- to high-grade and eventually may become invasive carcinoma[34]. While the inactivation of APC acts as an initiating event and p53 mutation was found to be a relatively late step in sporadic CRC, IBD-associated CRC shows the reverse pattern. Loss of p53 function can prolong NF-κB activity and promote chronic inflammation, and NF-κB activity is particularly abundant in inflamed colonic tissues[15,35]. Although the precise molecular mechanisms remain unclear, many studies have indicated that inhibiting NF-κB expression impedes cancer progression, suggesting that modulating NF-κB activity in inflammatory disease is crucial. Our present study provides a molecular link between ITF2 and NF-κB in the context of colon cancer development.

The canonical NF-κB pathway functions as a rheostatic transcription factor, and sustained activation leads to inflammation and epithelial injury in the intestine. In this study, we suggest that p65—not p50—directly binds to ITF2 and inhibits the degradation of ITF2 by suppressing ubiquitination. Similarly, Yadi et al. revealed that Snail, the master transcription factor for the epithelial-to-mesenchymal transition (EMT), is stabilized by the inflammatory cytokine TNF via activation of the NF-κB pathway and contributes to inflammation-induced cell migration and invasion[36]. In addition, NF-κB-mediated stabilization of Slug, which is also known as Snail2, underlies the inflammation-induced EMT and metastasis in head and neck squamous cell carcinoma[37]. Taken together, NF-κB could stabilize the key proteins that are responsible for cancer initiation and development, thereby affecting cancer development.

There is an inconsistency that whether the mRNA levels of ITF2 are upregulated in UC/CD patients. Some of the previous reports performed microarray analysis using colon mucosal biopsies and showed increased ITF2 mRNA levels in UC and CD patients relative to a healthy individuals[23,24,38]. In addition, Noble et al. presented upregulated ITF2 mRNA expressions with a positive correlation with TNF levels in UC tissues[22]. On the other hand, Fig. 2A, B, and many of the other GEO datasets in Fig. 2I exhibited no significant changes in ITF2 mRNA levels across the samples. We might suspect that this conflict result would be coming from the inconsistent sample collection method from the patient tissues. The papers mentioned above mostly utilized biopsies from the colonic mucosa derived from healthy, UC, or CD patients which enable possible contamination of lymphocytes. Besides, there is a chance that samples might be collected in the randomly selected areas, which means the ratio of the lymphocytes and epithelial compartment could be arbitrarily decided. Thus, if the RNAs were extracted from the colonic mucosa, and not even sorted, mucosa samples could randomly contain lymphocytes, making it hard to interpret the data. That is why many of the sequencing data showed contradictory results. However, our in vitro data (Fig. 2A, B) and GSE 11223 dataset which was done in epithelial biopsies showed that the ITF2 mRNA levels were comparable.

TNF has long been implicated in the development of colitis, and indeed, the pharmacologic blockade of TNF with monoclonal

antibodies has shown great efficacy in the treatment of IBD patients[39]. In addition, TNF has also been associated with some other epithelial malignancies, particularly skin cancer, in mice, implying that TNF may play a similar tumor-accelerating role in CAC[40]. Abrogation of TNF signaling or deficiency of TNF-Rp55 in mice led to reduced colitis features[41]. Indeed, TNF is known to be a major cytokine in CAC patients[42]. Therefore, we simply used this cytokine to mimic inflammatory conditions in vitro as an alternative to DSS in vivo. Our data demonstrated that TNF treatment successfully induced ITF2 expression in both in vitro and ex vivo conditions (Figs. 1–3).

Ubiquitylation arbitrates various cellular processes, such as endocytosis, the cell cycle, and protein degradation[43]. In this process, E3 ligases are a critical component of the ubiquitin-proteasome system, determining the substrate specificity of the cascade. Although over 600 putative E3 ligases have been reported, many of their functions are poorly identified[44]. Here, we identified that Parkin can act as a E3 ligase for ITF2 ubiquitination, suggesting that Parkin plays a role in the context of colon cancer development. Parkin dysfunction is highly associated with Parkinson's disease and other neurodegenerative diseases, including Alzheimer's disease[45]. However, a growing body of evidence implicates Parkin in tumorigenesis through regulation of the cell cycle, cell proliferation, apoptosis, and metastasis[46]. A previous report demonstrated that Parkin targets HIF-1α for ubiquitination and induces degradation to inhibit breast tumor progression[47]. Because E3 ubiquitin ligase can recognize substrates that contain a specific motif or sequence, it is reasonable that an E3 ligase such as Parkin could be used to identify several substrate candidates, such as HIF-1α and ITF2.

In the present study, we used an AOM/DSS murine model to induce CAC in mice. AOM is a colonic genotoxic carcinogen that is widely used for the assessment of colorectal carcinogenesis in rodents[33]. Although AOM itself does not induce dysplasia or cancer, in combination with repeated cycles of DSS, it significantly increases the incidence of dysplastic lesions and tumors in the intestinal epithelium in a manner reminiscent of human CAC[33]. Because of its high reproducibility and potency, this model has been utilized to explore interactions between inflammation and cancer in the colon. We found that ITF2 inhibition facilitated NF-κB translocation to the nucleus and upregulated the expression of pro-tumorigenic cytokines such as IL-6, TNF, IL-1β, and IL-8; thus, these results led us to hypothesize that chronic inflammation without AOM might induce tumor formation in ITF2 conditional knockout mice. Surprisingly, we revealed that DSS-induced chronic inflammation was sufficient to elevate the number of dysplasia or microadenoma-like lesions in the colonic intestine (Supplementary Fig. S13).

We also compared the expression of key molecules such as ITF2, p65, and β-catenin in tissues of patients with sporadic CRC and CAC. Mutations affecting β-catenin, axin, and APC, the key components of Wnt signaling, have been reported to be the main drivers of sporadic CRC progression[42]. In line with this finding, we also observed that β-catenin activity was already increased at the adenoma stage, and was further augmented in carcinoma tissue (Supplementary Fig. S14),

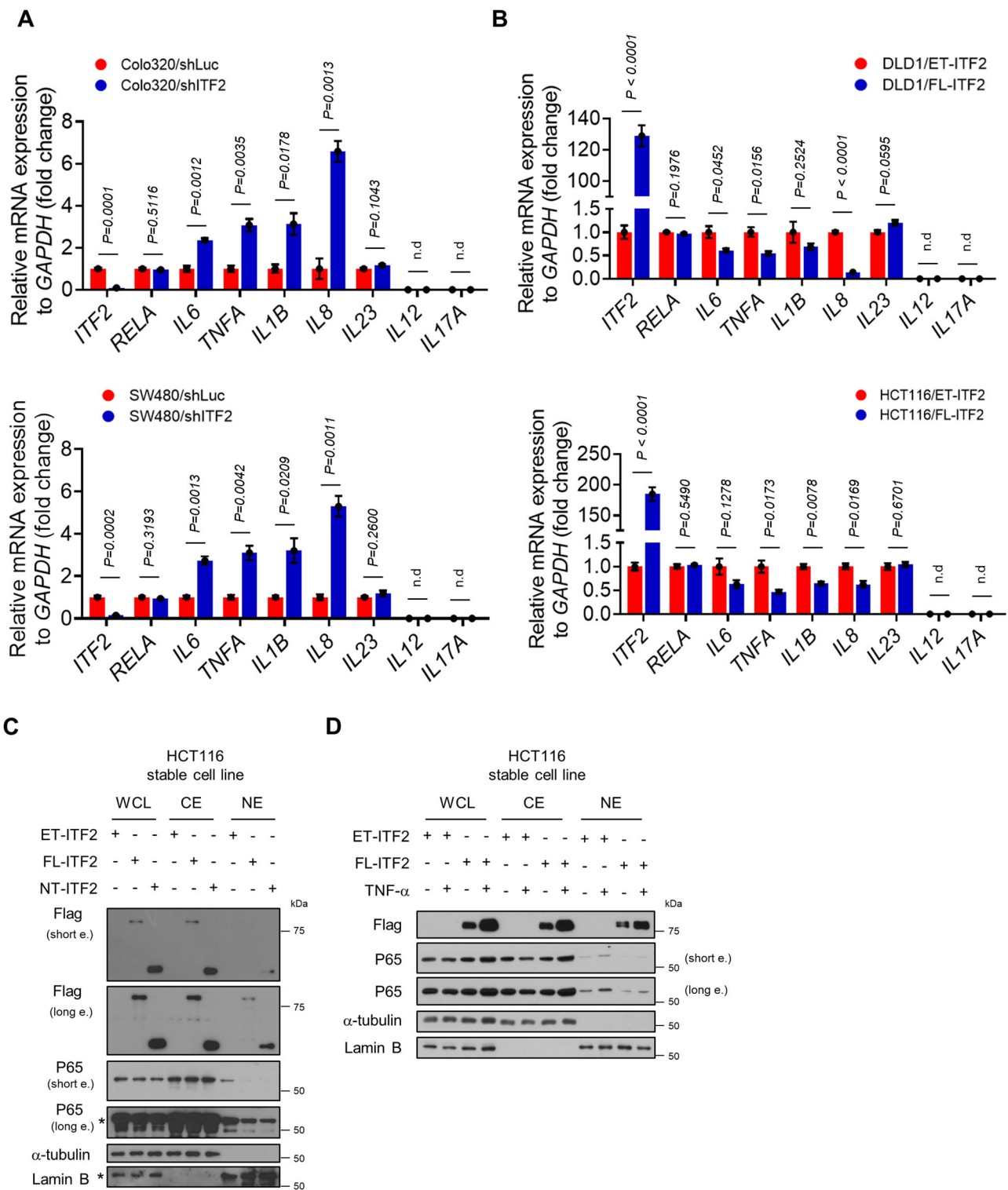

**Fig. 5 | ITF2 binds to p65 and interrupts p65 translocation into the nucleus, resulting in reduced TNF-induced p65 activation. A**, **B** Real-time qRT-PCR of key pro-tumorigenic cytokines related to CAC development in stable colon cancer cell lines transfected with the indicated plasmids (*n* = 3, biologically independent experiments, exact *P* values are shown in the figures). **C** The nuclear fraction experiment in HCT116 stable cell lines. Plasmids encoding ET-ITF2 (empty), FL-ITF2 (full length), or NT-ITF2 (N-terminal) were transfected into HCT116 cells and selected using G418 to generate stable cell lines. *n* = 2, biologically independent

experiment. **D** The HCT116 stable cell lines which were transfected with ET-ITF2 or FL-ITF2 were treated with or without TNF for 8 h, and then cells were fractionated into cytoplasmic and nuclear extracts. *n* = 2, biologically independent experiment. **C**, **D** The extracts were immunoblotted by the indicated antibodies (**C**, **D**). In all immunoblot analyses, the data are representative of three independent experiments. All results are presented as means ± SEM. Statistical significance was determined by the unpaired two-tailed Student *t* test (**A**, **B**). ND not detected. Source data are provided in the Source Data file.

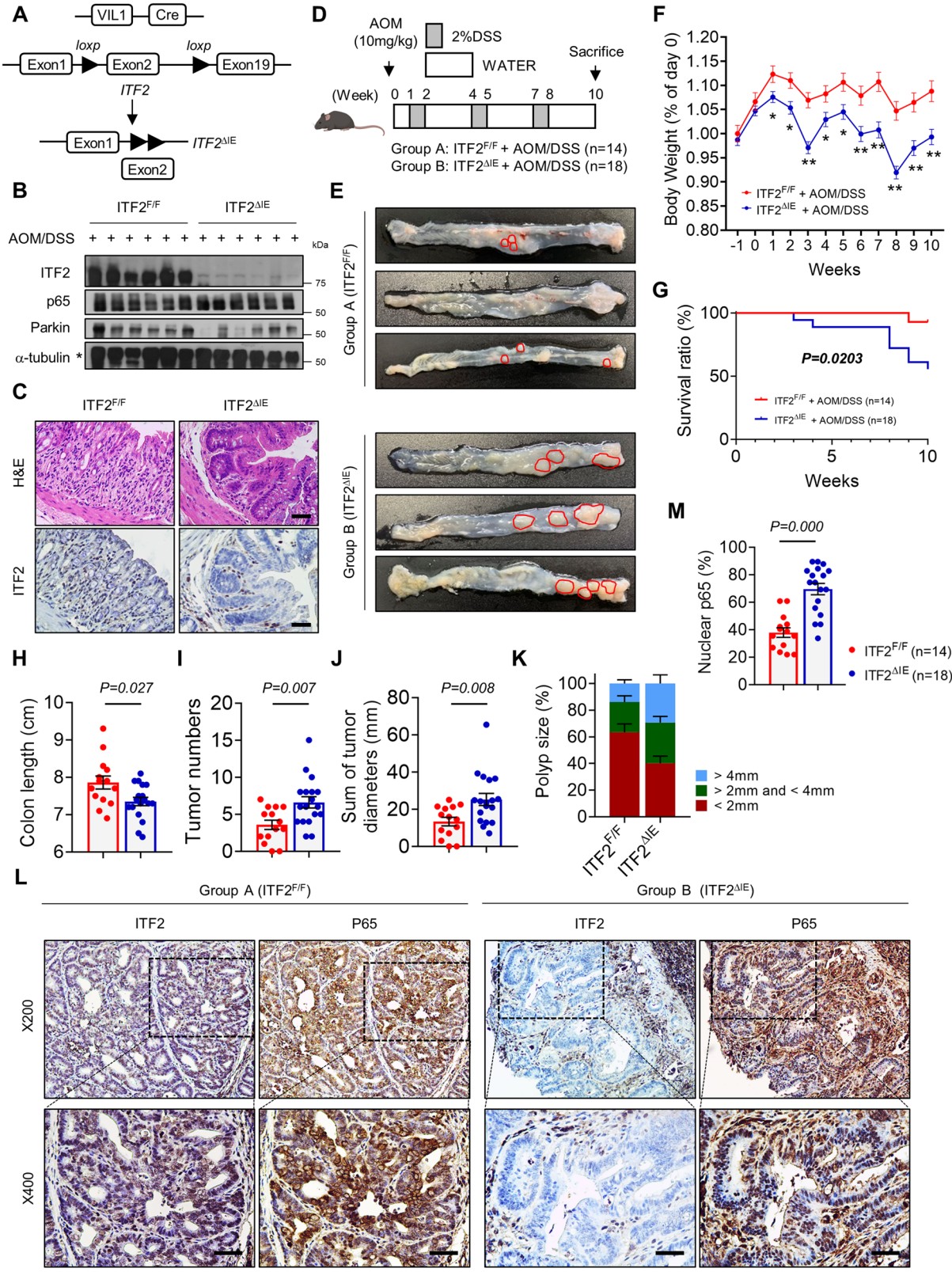

whereas p65 levels were not substantially different across the tissues. Considering that β-catenin interacts with NF-κB and inhibits its activity, these results seem to be plausible[48]. In the CAC tissues, β-catenin activity was rarely detectable in dysplastic conditions (around 10%) but was prominent in carcinoma samples (Fig. 7). This implies that key factors other than β-catenin could drive colitis-induced colon cancer malignancy. The NF-κB activity was significantly increased throughout the CAC tissues. Cooks et al. reported that a p53 mutation is an early event in tumor development in CAC, unlike sporadic CRC, where it commonly occurs in later stages, such as the adenoma-to-carcinoma transition; furthermore, loss of the p53 mutation in IECs significantly increased NF-κB activity, suggesting that the NF-κB pathway or NF-κB-driven molecules can mediate cancer progression[15]. Our results show that p65, a key subunit of NF-κB, directly interacts with ITF2 and

**Fig. 6 | ITF2 loss exacerbates AOM/DSS-induced tumor development.**
**A** Schematic representation of mice with temporal disruption of intestinal epithelial ITF2 (*ITF2*^F;F;VilCre^) using Cre recombinase driven by the villin promoter.
**B** Western blot analysis of ITF2, p65, and Parkin in crypt epithelial cells isolated from *ITF2*^ΔIE^ mice and littermates treated with 3.5% AOM/DSS (*n* = 6 for WT, 6 for KO respectively, biologically independent animals.). **C** Representative hematoxylin and eosin (H&E) and immunohistochemical staining for ITF2 in colon sections. Five or more mice were stained, with similar results. **D** Schematic representation of AOM/DSS-induced colon tumorigenesis performed in *ITF2*^ΔIE^ mice and littermates. Mouse icon was created with BioRender (BioRender.com). **E** Representative gross macroscopic images of AOM/DSS-induced colorectal tumors in control littermates (3 out of 14 mice; top) and *ITF2*^ΔIE^ mice (3 out of 18 mice; bottom). Red circles denote the edge of the colonic tumor. **F** Body weight (*ITF2*^F/F^, −1w–8w, *n* = 14; 9w–10w, *n* = 13 and *ITF2*^ΔIE^; −1w–3w, *n* = 18; 4w, *n* = 17; 5w–8w, *n* = 16; 9w, *n* = 13; 10w, *n* = 10; biologically independent animals; 1w, *P* = 0.0241; 2w, *P* = 0.0110; 3w, **P* = 0.0002;

4w, **P* = 0.0290; 5w, **P* = 0.0188; 6w, ***P* = 0.0062; 7w, ***P* = 0.0023; 8w, ***P* = 0.0001; 9w, ***P* = 0.0013; 10w, ***P* = 0.0049) (**G**) the survival rate of the AOM/DSS-induced CAC mouse model were monitored. Kaplan–Meier survival rate analyses were followed up until 10 weeks after AOM/DSS delivery. The dot plots of colon length (**H**), colonic tumor numbers (**I**), and the sum of tumor diameter (**J**). **K** Polyp size distribution (*ITF2*^F/F^, *n* = 14; *ITF2*^ΔIE^, *n* = 18). **L** Representative immunohistochemical images for ITF2 and p65. Areas indicated with squares are shown as magnified images. The data are representative of three independent experiment. **M** The number of nuclear-positive p65 cells was counted and compared in each group of mice. Scale bars, 100 μm. Individual values are indicated by dots (*ITF2*^F/F^, *n* = 14; *ITF2*^ΔIE^, *n* = 18 for **H, I, J, K, M**, biologically independent animals). *P* value was calculated by log-rank test (**G**). All results are presented as means ± SEM. Statistical significance was determined by the two-tailed Mann–Whitney *U* test for pairwise comparisons (**F, H, I, J, K, M**). Source data are provided in the Source Data file.

---

prolongs ITF2 expression by decreasing ubiquitination, implying that ITF2 can mediate CAC progression.

Our study has identified a potential tumor suppressor gene, ITF2, which may play a role in inhibiting inflammation-mediated cancer progression. Based on the previous microarray dataset (Fig. 7C), UC-associated neoplastic legions, UC-associated carcinoma tissues, and sporadic CRC tissues displayed rare ITF2 mRNA expressions, suggesting that the levels of ITF2 have already disappeared in the mRNA levels. We acknowledge that there are limitations to our findings, such as the lack of validation on the specific factors or genetic mutations that could be involved in downregulating ITF2 expression during the dysplasia-to-carcinoma transition. Therefore, further studies are needed to examine the molecular details of our findings on a larger scale, including genetic mutations in ITF2, SNPs in specified genes, and post-translational modification of ITF2.

## Methods
### Human tissue specimens
Human CAC tissue specimens and sporadic CRC specimens were obtained from the Seoul National University Hospital Tissue Bank and Boramae Hospital Tissue Bank. Patient's medical records were reviewed, and tissues from all potential cases of CAC and CRC were collected. A pathologist (J.H.K.) reviewed all slides from all cases to check for evidence of colitis and carcinoma, and a gastroenterologist (S.J.K.) reviewed the medical records for the clinical history of IBD. They then selected the proper tissue blocks for further analysis, and samples were classified into three groups: normal, dysplasia, and carcinoma. In terms of CRC, samples were classified into three groups: normal, adenoma, and carcinoma. Regarding the UC samples, tissues were classified into two groups: normal and dysplasia. All procedures were approved by the Institutional Review Board and Committee of Seoul National University Hospital (approval 1912-056-1087 and 2105-055-1218).

### Mice
All in vivo experiments were approved by the Institutional Animal Care and Use Committee (IACUC) of Seoul National University (approval SNU-180306-3-2, SNU-180306-2-3, and SNU-181214-4-3) and carried out in strict accordance with good animal practice as defined by the governmental and international guidelines of animal experiments. Genetically engineered C57BL/6 mice were used. *Vil1-Cre* mice (004586) were purchased from the Jackson Laboratory (ME, USA). The loxP-floxed ITF2 mice were generated by ToolGen Inc. (Seoul, South Korea). ITF2^fl/fl^ mice were crossed with mice that expressed Cre recombinase under the control of villin promoter to abrogate ITF2 expression in an intestinal epithelial cell (IEC)-specific manner. For the acute colitis model and colonic organoids, wild-type mice were utilized (Koatech, Gyeonggi-do, South Korea). The genotypes of the mice were performed by polymerase chain reaction, and the absence of ITF2

proteins was confirmed by immunoblotting (Fig. 6B). Both male and female mice aged 6–8 weeks were used at the time of experiments unless indicated otherwise. Sample sizes for mouse experiments were empirically determined, and mice were randomly allocated to the control or experimental groups. Animal studies were conducted in a gender- and age-matched manner using littermates for each experiment. All mice were of C57BL/6 background. All mice used in this study were housed at the specific-pathogen-free (SPF) animal facilities with a controlled temperature and free access to food and water under a 12 h light: dark cycle. To avoid bacteria infection & mycoplasma contamination, and maintained the mouse condition similarly as much as we can, we used the autoclaved chow diet and drinking water offered by the animal facility.

### Generation of ITF2^fl/fl^ mice
ITF2^fl/fl^ mice were produced by ToolGen Inc (Seoul, South Korea). CRISPR/Cas9 were designed to target the intron 1 and 18 of the ITF2 gene, respectively. The guide RNA (gRNA) sequences used in this study were: 5′-CCTAAACCAAAATATCACCT-3′ for intron 1 and 5′-CTCTTAACGGACCTATGCCC-3′ for intron 18. The synthesized single-stranded oligonucleotides (ssDNA) were 5′-catttctcgctctacctcctgttgca taaattgatgcgctgaagagcggcttaccccccttctttcctggatttgATAACTTCGTATA ATGTATGCTATACGAAGTTATgttttatagtggatattactagtttgatcgtctctttgg aaatacaatatcgctattttt-3′ and 5′-tcagcctggagcagcaagttcgaggtcagcagag agatgcttaagcaccctcccgagaattgcctggaATAACTTCGTATAATGTATGCT ATACGAAGTTATtacgggaccgtatcaaactctgtctcccgtaattaacctggagaaga ttcttctaagtag-3′. These ssDNAs were used as donor DNAs to insert LoxP site at the intron 1 and 18 of ITF2 gene, respectively. Then, ssDNAs were co-injected into fertilized wild-type oocytes together with Cas9 protein and gRNAs. Microinjection of one-cell embryos was achieved as previously described[49,50]. To screen founders carrying both LoxP sites in the ITF2 gene, we performed PCR assay using genomic DNA derived from pubs that were generated from the microinjected embryos. The genomic regions spanning each LoxP site were amplified by PCR. The primer sequences used in PCR were: F: 5′-TTGAGACCGG TTCGTGCAGT-3′ and R: 5-GGAAGGAACTGGACAAATGTTGGG-3′ to identify LoxP in the intron 1, F: 5′-AGCTCCACCTGAAGAGCGAC-3′ and R: 5′-AGGTTGGCCAATCATGTCTGA-3′ to identify LoxP in the intron 18. The candidates were cloned in T-Blunt PCR Cloning Vector (Solgent Co. Ltd., Daejeon, South Korea), and were validated by direct sequencing analysis. A lack of ITF2 expression in crypt epithelial cells from homozygous mice was confirmed by immunoblotting (Fig. 6B).

### Induction of tumorigenesis
To explore tumorigenesis, the following two murine models were used: the azoxymethane–dextran sulfate sodium (AOM-DSS) model of CAC and the chronic DSS colitis model. ITF2^fl/fl^Vil1-Cre (cKO) and ITF2^fl/fl^ (control) littermates were co-housed after weaning. For the AOM-DSS or chronic DSS model, 8-week-old ITF2^fl/fl^ and ITF2^ΔIE^ mice were

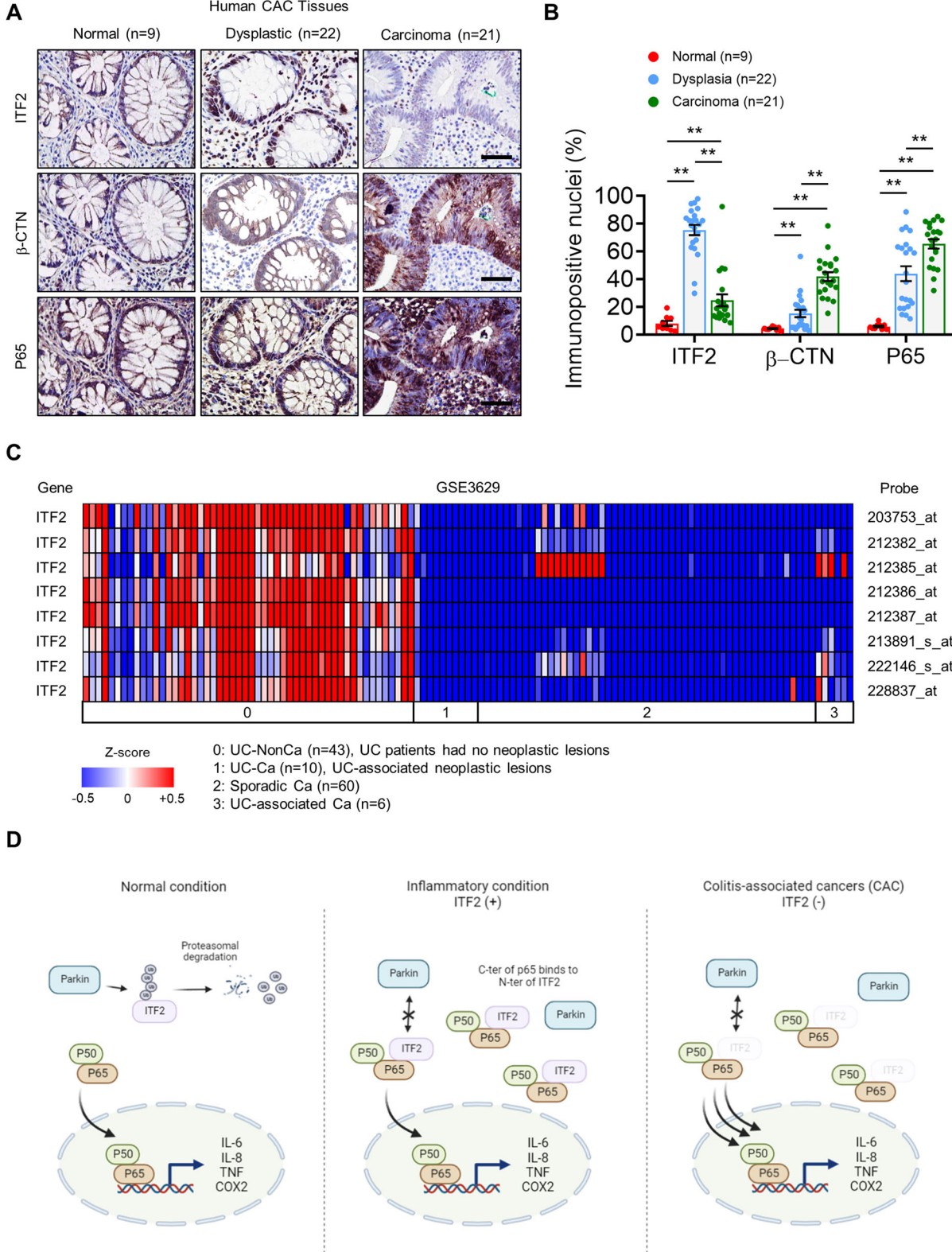

intraperitoneally injected with or without AOM (10 mg/kg body weight) (Sigma-Aldrich, MO, USA) on day 1, and then cycled on and off, starting from day 7, 2% DSS (MP Bio, OH, USA; MW 36,000 to 50,000) in their drinking water. In the acute colitis model in WT mice, mice received water with 3.5% DSS for 7 consecutive days and were then sacrificed. For the crypt epithelial cell isolation, mice were sacrificed on days 0, 3, 5, and 7 respectively. During the DSS cycles and the recovery phase, weight loss, stool consistency, and stool occult blood were monitored daily. After AOM-DSS or DSS administration, the mice were sacrificed and their colons were removed to measure colon length. Subsequently, the colons were opened longitudinally, and the size and number of tumors in each colon were examined.

**Fig. 7 | Negative expression of ITF2 and p65 in CAC tissues. A** Representative immuno-stained images of ITF2, β-catenin, and p65 in normal, dysplasia, and carcinoma CAC tissues. The data are representative of three independent experiments. **B** Nuclear-positive expression levels of ITF2, β-catenin, and p65 were counted and analyzed in CAC specimens. **C** The GEO dataset (GSE3629) was analyzed for ITF2 gene expression in UC-NonCa (43 patients), UC-Ca (10 patients), sporadic Ca (60 patients), and UC-associated Ca (6 patients). The heat-map image was generated using the Multiple Experiment Viewer (MEV) software. The color scale represents relative expression values; blue, low expression score; red, high expression score. **D** Graphical summary of ITF2 regulation by p65. Schematic illustrations were generated with BioRender (BioRender.com). Normally, ITF2 is ubiquitinated by a Parkin, leading to ubiquitination and proteasomal degradation of ITF2. In inflammatory conditions, such as dysplasia, where p65 is increased, p65 levels are upregulated, leading to ITF2 protein stabilization, thereby interrupting p65 activation. In carcinoma, p65 is further activated owing to ITF2 loss and leads to excessive expression of pro-tumorigenic cytokines such as IL-6, TNF, IL-8, and IL-1β, thereby contributing to eventual CAC development. Scale bars, 100 μm. All results are presented as means ± SEM. Statistical significance was determined by Kruskal–Wallis tests followed by the two-tailed Mann–Whitney $U$ test (**B**, **P < 0.0001) for pairwise comparisons. Source data are provided in the Source Data file.

### Collection of tissue samples

Mice were euthanized by cervical dislocation under avertin (Acros Organics, NJ, USA) anesthesia and the colon was removed, flushed, cut wide open, and pinned flat on a Petri dish (SPL Life Sciences, Gyeonggi-do, South Korea). The half of the colon (proximal part) was taken as a tissue sample for subsequent protein analysis by isolating crypt epithelial cells from tissues. The other half (distal colon) was fixed with 4% PFA (Biosesang, Seongnam, South Korea) for paraffin embedding and histological analysis. Induced tumors were evaluated histologically by H&E staining and subsequent IHC staining. The clinical course of the disease was traced by every other day by measuring the body weight of mice.

### Antibodies and reagents

Information about the antibodies and reagents used in this study is provided in Tables S1–2.

### Protein-protein binding sites prediction

Alphafold2 predicts three-dimensional (3D) protein structures based on the protein sequences of the ITF2 and Parkin, and then showed promising Lys residues that are supposed to be ubiquitinated by the Parkin in the N-terminus of the ITF2[51].

### Cell lines and cell culture

Human embryonic kidney (HEK293T), and human colon cancer cell lines (Colo320DM, SW480, HCT116, LoVo, DLD-1, LS174T, HT-29, SNU-283, HCT-15, SNU-1033, CaCo2, WiDr, and Colo205) were obtained from the American Type Culture Collection (VA, USA) and/or Korean Cell Line Bank (Seoul, South Korea). The cell lines were authenticated by the supplier based on growth, morphology, and STR (short tandem repeat) DNA profiling of the cell line was analyzed in Korean Cell Line Bank (Seoul, South Korea). All cell lines were routinely tested to confirm the absence of mycoplasma, using the MycoAlert Plus Mycoplasma Detection Kit from Lonza (Walkersville, MD, USA). The cell lines were maintained in RPMI-1640 or DMEM supplemented with 10% heat-inactivated FBS and 1% penicillin-streptomycin in a 5% $CO_2$ humidified incubator at 37 °C. Colo320DM and SW480 colon cancer cells with stable knockdown of ITF2 by short hairpin RNA (shRNA) were established previously[20], and passaged using puromycin (10 μg/ml). HCT116 colon cancer cells were previously transfected with Flag/streptavidin binding protein-ITF2_FL (full length), Flag/streptavidin binding protein-ITF2_NT (N-terminus only), or empty Flag/streptavidin binding protein vector, and passaged using G418 (1 mg/ml)[20].

### Cycloheximide chase assay

Cells were cultured with TNF (20 ng/mL) for 8 h followed by cycloheximide (CHX; 50 μM) treatment and harvested at indicated time points. Cells were lysed with sample buffer and subjected to western blot analysis. Protein levels were evaluated with densitometric intensity.

### Cytokine stimulation of colon cancer cells or organoids

Colonic organoids and colon cancer cells were stimulated with TNF (Peprotech, NJ, USA) in a concentration- and time-dependent manner, as indicated in each figure. In some cases, cells were pre-incubated with medium containing small molecule inhibitors such as rapamycin (5 μM), PD98059 (20 μM), SB203580 (10 μM), AG490 (50 μM), MK2206 (5 μM), BAY 11-7082 (10 μM), TPCA1 (20 μM), BAY 11-7085 (10 μM) or MG132 (20 μM) for 1 h followed by TNF (20 ng/mL) stimulation.

### Expression dataset and protein-protein interaction analysis

An analysis was conducted of the available GSE microarray datasets ("GSE38713", "GSE11223", "GSE37283", "GSE112366", "GSE75916", "GSE1710", "GSE20881", "GSE87473") with associated pathological stages for each sample (normal/UC, uninflamed in UC/inflamed in UC, normal/CD, or epithelial organoids from non-IBD/epithelial organoids from UC) to compare ITF2 mRNA levels. The values of the 315135_at probe (corresponding to ITF2) for each group were used and calculated. ITF2 mRNA expression levels were compared in each group of patients and visualized using the Multiple Experiment Viewer software with the GSE 3629 dataset. The list of ITF2-interacting E3 proteins was examined from the BioGRID database[52].

### Fractionation of cytoplasmic and nuclear components

HCT116 stable cell lines established with small hairpin RNA targeting ITF2[20] were used and treated with or without TNF (Peprotech, Rocky Hill, CT, USA). Cells stimulated or not with TNF were centrifuged at $800 \times g$ for 5 min and gently homogenized in a hypotonic solution containing 20 mM Tris/HCl (pH 7.8), 1.5 mM MgCl2, 10 mM KCl, 0.2 mM EDTA, 0.5% NP-40, 0.5 mM dithiothreitol, and 0.5 mM phenylmethylsulfonyl fluoride. The cell lysates were centrifuged at $3000 \times g$ for 10 min at 4 °C, and the supernatant was collected as the cytosolic fraction[53].

### Immunoblotting and immunoprecipitation

Cell lysates were electrophoresed on sodium dodecyl sulfate-polyacrylamide gels and transferred to Immobilon-P (Millipore, MA, USA). Membranes were incubated sequentially with primary antibodies and horseradish-peroxidase–conjugated secondary antibodies and visualized using an electrochemiluminescence kit (Thermo Scientific, IL, USA). To analyze protein interactions, cell lysates were incubated with antibodies and precipitated with protein A/G Sepharose beads (GE Healthcare, Buckinghamshire, UK)[54]. For immunoblotting, protein samples were boiled in a sample buffer for 5 min at 94 °C. Denatured proteins were electrophoresed on SDS-PAGE gels and transferred to Immobilon-P membranes (Millipore, MA, USA). Membranes were blocked with a Tris/saline solution containing 5% low-fat milk for 1 h at room temperature and incubated with indicated primary antibodies overnight at 4 °C. The blocked membranes were washed with Tris-buffered saline containing 0.05% Tween 20 three times for 15 min each time. The horseradish peroxidase (HRP)–conjugated secondary antibodies (Invitrogen, CA, USA) were applied for 1 h at room temperature, and bands were visualized using an Amersham ECL Plus Western Blotting Detection reagent (GE Healthcare, Buckinghamshire, UK). For immunoprecipitations, cell lysate was prepared using binding buffer (50 mM Tris-HCl pH 7.5, 150 mM NaCl, 1 mM EDTA, 1% Triton X-100, and 5% glycerol) containing protease inhibitor

cocktail (Sigma-Aldrich, MO, USA) and phenylmethylsulfonyl fluoride (PMSF, Sigma-Aldrich, MO, USA). Cell lysates were cleared by centrifugation, and incubated with 1 mg of indicated antibodies and A/G agarose beads (Santa Cruz Biotechnology, CA, USA) on a rotator for 6 h at 4 °C. The immunoprecipitates were then washed four times and eluted with a sample buffer (125 mM Tris-HCl pH 6.8, 4% SDS, 20% glycerol, 10% 2-mercaptoethanol, 0.004% bromphenol blue) or eluted by peptide competition. When c-Myc–tagged proteins needed to be immunoprecipitated, 2 mg protein of cell lysates were incubated with anti-c-Myc affinity gel (Sigma-Aldrich, MO, USA) for 4 h, and eluted with c-Myc peptides (Sigma-Aldrich, MO, USA). For the immunoprecipitation of FLAG-tagged proteins, 2 mg protein of cell lysates were incubated with anti-FLAG M2 affinity gel (Sigma-Aldrich, MO, USA) for 4 h, and eluted with 3X FLAG peptides (Sigma-Aldrich, MO, USA). For HA-tagged proteins, 2 mg protein of cell lysates were incubated with anti-HA affinity gel (Sigma-Aldrich, MO, USA) for 4 h, and eluted with HA peptides (Sigma-Aldrich, MO, USA). The mouse and rabbit purified IgG was used as a negative control. The uncropped blots were presented in the Source data file.

## Immunohistochemistry and visualization

The formalin-fixed paraffin-embedded sections (4 μm) of human CAC tissues, sporadic CRC tissues, or mice colonic tissues were rehydrated and autoclaved at 121 °C for 10 min in 100 mM citrate buffer (DAKO, CA, USA) for epitope retrieval. The sections were treated with methanol-diluted 3% hydrogen peroxide for 10 min followed by the 3% bovine serum treatment at room temperature for at least 1 h to block nonspecific bindings. They were incubated with antibodies against ITF2 (1:100), β-catenin (1:100), and p65 (1:100) overnight at 4 °C. Then, the slides were incubated with Polink-2 plus HRP detection system (GBI labs, Mukilteo, WA) and visualized by DAB. The mouse and rabbit IgG was used as a negative control. (Santa Cruz Biotechnology, CA, USA). To standardize color development, the DAB incubation time was fixed in all experiments. All immunostained sections were counterstained with Meyer's hematoxylin (Vector Labs, CA, USA), dehydrated in graded ethanol, cleared with xylene, mounted with VectaMount permanent mounting solution (Vector Labs, CA, USA), and then examined under a bright-field microscope (ECLIPSE Ci-L; Nikon, Tokyo, Japan). We selected three different high-power fields (HPFs) per patient. Two examiners blinded to the experimental group determined and averaged the expression levels. When the examiners disagreed, a consensus was reached by reviewing the specimens at a multi-head microscope by our research team. To calculate the percentage of target proteins in the human specimens and mice colonic tissues, immunopositive nuclei cells in each high-power field (HPFs) were counted and shown as a percentage (per 100 cells)[55,56].

## Immunofluorescence and confocal microscopy

Isolated crypt epithelial cells were fixed in 4% PFA (Biosesang, Seoul, South Korea) for 20 min at room temperature. Fixed cells were permeabilized for 30 min at 37 °C with 0.2% Triton X-100 in PBST (PBS + 0.05% Tween 20), and then blocked for 2 h at room temperature with 3% bovine serum albumin (BSA) in PBST. For immunofluorescence of organoids, organoids were cultured with Dispase (Invitrogen, CA, USA) for 20 min at 37 °C, and fixed with 4% PFA for 30 min. Then, organoids were permeabilized with PBS containing 0.2% triton X-100 and blocked with a blocking solution containing 5% BSA, 3% normal goat serum, and 0.2% Triton X-100 for at least 2 h at room temperature. Cells or organoids were incubated overnight at 4 °C with primary antibodies against E-Cad (1:100), CD45 (1:100), and Ki-67 (1:100). After several washes with PBST, cells were stained with the Alexa Flour 488 IgG anti-rabbit (for E-Cad and Ki-67) and Alexa Flour 555 IgG anti-mouse (for CD45) in the dark for 1 h at room temperature. Finally, they were mounted to slide with VECTASHIELD mounting solution (Vector Labs, CA, USA) with DAPI (Sigma-Aldrich, MO, USA), and imaged. The fluorescence images were taken using a confocal laser scanning microscopy (Carl Zeiss, Jena, Germany).

## In vivo and in vitro ubiquitination assays

For in vivo ubiquitination assays, cells were transfected with the indicated vectors which were described in each figure for 48 h. Then, cells were cultured with MG132 (10 μM) for 8 h before harvest, and the levels of target protein ubiquitination were determined by IP with an α-Myc, α-Flag, or α-ITF2 antibodies respectively followed by immunoblot assays with an α-HA antibody. To perform in vitro ubiquitination assays, 7 μl of purified F/S-ITF2 and Myc-Parkin proteins from HEK293 cells were added to a reaction with 400 ng of E1 (UBE1; Enzo Life sciences, PA, USA), 800 ng of E2 (UbcH5a; Boston Biochem, MA, USA), and 1 μg of ubiquitin (Sigma-Aldrich, MO, USA) in 25 μl of reaction buffer (40 mM Tris (pH 7.6), 50 mM NaCl, 5 mM MgCl2, 1 mM dithiothreitol, 2 mM ATP) for 3 h at 37 °C. The reaction was terminated by the addition of 4× sample buffer and boiling. Sequentially, immunoblotting was performed using an α-Flag antibody to detect ubiquitin-conjugated ITF2 proteins.

## IEC isolation and culture of colonic organoids

Mouse colonic epithelial cells were isolated, and organoids were generated and cultured with slight modifications[57–59]. About 10 cm of colon tissues were opened longitudinally, washed with cold Hank's balanced salt solution (HBSS; Gibco, CA, USA), chopped into around 5 mm pieces, and incubated in 10 mM EDTA (Sigma-Aldrich, MO, USA) with HBSS for 45 min on ice without shaking. After removal of the EDTA medium, the tissue fractions were vigorously shaken to isolate colonic epithelial cells and filtered through a 70 μm cell strainer (BD Falcon, CA, USA) to discard residual villous material, and centrifuged at 400 g for 5 min to gather crypts. Crypt fractions were re-suspended with complete growth medium [advanced DMEM/F12 supplemented with B27, N2, EGF, HEPES, Glutamax, N-acetylcysteine] and purified by successive centrifugation steps (400 g, 1 min). The final fraction of crypts was used for immunoblotting, immunofluorescence, and organoids generation. For generation and culture of organoids, purified crypts were counted and cultured in WENR medium [Wnt-3, EGF, Noggin, and R-spondin1] with 50 μl mixture of growth factor-reduced Matrigel (BD Bioscience, CA, USA) and complete growth medium (at a ratio 2:1)[58]. The medium was refreshed every other day and maintained until day 14. Phase contrast images were taken on the indicated days (Fig. 1B). Organoids were stimulated with recombinant mouse TNF in a concentration-dependent manner for 8 h. For passage, organoids were mechanically disrupted, and transferred to fresh Matrigel[59].

## FACS analysis of cell death

The effect of ITF2 on TNF-mediated cell death pathways such as apoptosis and necrosis was examined by the combined application of Annexin V-FITC Apoptosis Detection Kit (Invitrogen, USA) followed by flow cytometric analysis according to the manufacturer's instructions. Briefly, the Colo320DM cells were transfected with Myc-ITF2 followed by the TNF stimulation for 8 h, and 24 h respectively. The cells were resuspended in a binding buffer and then stained with FITC-conjugated annexin V antibody for 15 min in the dark at room temperature, and then incubated with propidium iodide sequentially. Flow cytometric analysis was immediately performed in a BD FACSDiva v8.0.1 (BD Biosciences, USA). The viable cells were located in the lower left quadrant (annexin V−/PI−). Early apoptotic and necrotic cells were sorted in the lower right (annexin V + /PI−) and the upper left (annexin V−/PI + ) quadrant, respectively. Late apoptotic cells were located in the upper right quadrant (annexin V + /PI + ).

## In vitro PLA assay

Duolink fluorescent reagents were purchased from Sigma-Aldrich (MO, USA) and performed as specified by the manufacturer.

## Luciferase assays

To monitor NF-κB luciferase activities, DLD-1 and HT-29 cells were co-transfected with 1 µg of pcITF2 plasmid, 1 µg of κB-luciferase plasmid, and 1 µg of β-galactosidase (β-gal) plasmid using Mirus TransIT-X2 (Mirus Bio, WI, USA) reagent. Colo320DM and CaCo2 cells were co-transfected with 1 µg of κB-luciferase plasmid, 1 µg of β-gal plasmid, or 60 nM siRNA targeting β-catenin using Mirus TransIT-X2 reagent. The final concentrations of DNA or siRNA were adjusted with pcDNA or control siRNA. After being stabilized for 48 h, cells were treated with human recombinant TNF in a time-dependent or concentration-dependent manner before being harvested for assays. β-gal expression was used as an internal control to normalize the transfection efficiency. To measure the transcriptional activity of TCF4/β-catenin, Colo320DM, DLD-1, or SW480 cells were co-transfected with 500 ng pTOP (T-cell factor reporter plasmid)/FLASH, 500 ng pFOP (mutant T-cell factor reporter plasmid)/FLASH, 500 ng p65 and 500 ng β-gal vectors. In some cases, TNF was treated instead of p65 plasmid transfection. For all co-transfection assays, the total amount of DNA was kept constant with appropriate amounts of the respective control vectors. The final DNA concentrations were adjusted by adding pcDNA. Luciferase activity was analyzed using a Lumat LB960 luminometer (Berthold Technologies, Bad Wildbad, Germany) following the manufacturer's recommendations.

## Protein purification and in vitro binding assay

Plasmids for Glutathione-S-transferase (GST)-fused p65 peptides or His-tagged ITF2 peptides were introduced into *E. coli* strain BL21 (DE3) cells and the recombinant peptides were induced with 1 mM IPTG for 4 h at 37 °C. HEK293T cells were then transfected with mammalian expression plasmids and harvested. GST-tagged p65 peptides and His-ITF2 peptides were purified using Glutathione-Sepharose 4B beads (GE Healthcare, Buckinghamshire, UK) and Nickel-NTA resin (Qiagen, CA, USA), respectively. FLAG-tagged proteins were purified with FLAG affinity gels (Sigma-Aldrich, MO, USA), HA-tagged proteins with HA affinity gels (Sigma-Aldrich, MO, USA), and c-Myc-tagged proteins with c-Myc affinity gels (Sigma-Aldrich, MO, USA). Purified eluates were stored at −70 °C until use. For binding assays, 500 ng of GST-p65 peptides or FLAG-p65 peptides were mixed with the same amount of a His-ITF2 peptide or Myc-ITF2 respectively, and incubated in a binding buffer (25 mmol/L HEPES (pH 7.4), 150 mmol/L KCl, 12.5 mmol/L MgCl2, 10% glycerol, 1% NP-40, 0.1% bovine serum albumin, and 5 mmol/L dithiothreitols) for 1 h at 4 °C. The protein complex was pulled down with Glutathione beads for 2 h at 4 °C. Pull-downs were washed 4 times with the binding buffer and subjected to immunoblot analysis using α-GST and α−6xHis antibodies.

## RNA isolation and quantitative real time-polymerase chain reaction with reverse transcription

Crypt epithelial cells were isolated from the distal colon tissues. Total RNA was isolated using TRIZOL (Invitrogen, CA, USA), and sequentially cDNA synthesis was carried out using a Tetro cDNA Synthesis Kit (Bioline, London, UK). The quantitative real-time polymerase chain reaction was performed on 96-well optical plates containing SYBR Green Master Mix reagents (Enzynomics, Daejeon, South Korea) and primer mixtures. Data were analyzed using CFX Manager Software (Bio-Rad Lab, CA, USA), and the mRNA values of target genes were normalized to HPRT expression[60,61]. Polymerase chain reaction conditions were as follows: 10 min at 95 °C, 40 cycles for 15 s at 94 °C, 30 s at 50 °C, and 30 s at 72 °C, followed by a melt curve for 15 s at 95 °C, ramp to 60 °C for 1 min, followed by 20 s at 95 °C[20]. The nucleotide sequences of PCR primers are described in Table S4. The PCR efficiency in all runs was close to 100%.

## siRNAs, expression plasmids, and transfection

siRNAs were designed and purchased from IDT (Coralville, IA, USA), and plasmids (ITF2, β-catenin, ubiquitin) were constructed by reverse-transcription polymerase chain reaction and blunt-end ligation[62,63]. The NF-κB subunit (p65, p50) and PRKN plasmids were kindly provided by Dr. Hang-Rae Kim (Seoul National University College of Medicine) and Dr. Jongkyeong Chung (Seoul National University School of Biological Sciences), respectively. RelB and C-Rel plasmids were purchased from OriGene (Rockville, MD, USA). Site-specific deletion mutation of ITF2 was performed using PCR-based mutagenesis. The nucleotide sequences are summarized in Table S3. The complementary DNAs (cDNAs) of ITF2, β-catenin, and ubiquitin were cloned by reverse transcription and PCR using Pfu DNA polymerase, and the cDNAs were inserted into pcDNA, Myc-tagged, FLAG-tagged, HA-tagged, or FLAG/streptavidin-binding protein (SBP)-tagged vectors by blunt-end ligation. For transient gene silencing or protein expression, ~60% of confluent cells were transfected with small interfering RNAs or plasmids using the calcium phosphate method for HEK293T cells[64]. For other cells, Mirus TransIT-X2 (Mirus Bio, WI, USA) was used following the manufacturer's instructions. Cells were allowed to stabilize for 48 h before being used in the experiments.

## Statistical analysis

Statistical analyses were performed using SPSS software version 25.0 (IBM Corp., Armonk, NY, USA). Data were analyzed using the Student *t* test, the Mann–Whitney *U* test, Spearman correlation coefficients, and linear regression. All statistical tests were two-sided, and statistical significance was indicated as $^*P < 0.05$ and $^{**}P < 0.01$. Data are expressed as the mean ± standard deviation or the mean ± standard error of the mean. Illustrative figures were generated using Prism version 8.3.1 (GraphPad Software Inc., CA, USA) and SigmaPlot version 10.0 (Systat Software Inc., CA, USA). No statistical methods were used to pre-determine the sample size.

## Reporting summary

Further information on research design is available in the Nature Portfolio Reporting Summary linked to this article.

## Data availability

All data relevant to the study are included in the paper or uploaded as supplementary information. All data and sources associated with this study are available from the corresponding author upon reasonable request. The following reposited datasets of transcriptomic profiling of patients and healthy controls were accessed and analyzed for this manuscript: "GSE38713", "GSE11223", "GSE37283", "GSE112366", "GSE75916", "GSE1710", "GSE2088", "GSE87473". Source data are provided with this paper.

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

## Acknowledgements
This work was supported by a grant from the National Research Foundation of Korea (2020R1A4A2002903, 2022R1A2C2006075 and 2022R1A4A3034038, H.W.S).

## Author contributions
Conceptualization: M.G.L., H.W.S.; Methodology: M.G.L., H.W.S., S.H.L.; Investigation: M.G.L., Y.S.K., H.W.S., S.H.L., I.J.K., S.H.S. J.Y.K.; Animal Experiments: M.G.L., S.H.L. M.C.; Bioinformatic Analysis: M.G.L., S.H.S., I.J.K.; Formal analysis: M.G.L., Y.S.K.; Resources: H.W.S., J.H.K., S.J.K.; Writing—Original Draft: M.G.L., H.W.S.; Writing—Review and Editing: M.G.L., H.W.S.; Supervision: H.W.S., J.W.P.; Funding Acquisition: H.W.S. All authors have read and agreed to the published version of the paper.

## Competing interests
The authors declare no competing interests.
