## [Peer Review File · Nature Communications]

Protein stabilization of ITF2 by NF- κ B prevents colitis-associated cancer developmentREVIEWER COMMENTS

Reviewer #1 (Remarks to the Author):

The manuscript by Lee and colleagues shows that, on overexpression, the NF- κ B subunit RELA/p65 binds to the ITF2 transcription factor in intestinal epithelial cells, leading to ITF2 stabilization by preventing ubiquitination by the E3 ubiquitin ligase Parkin and proteasomal proteolysis. The authors propose that this mechanism is critical for inhibition of NF- κ B activation and transcription of NF- κ B-dependent proinflammatory genes in intestinal epithelial cells, thereby suppressing progression to CAC. Consistent with this view, intestinal epithelial-specific ITF2 deletion increased RELA/p65 nuclear translocation and colitis-associated carcinogenesis in the mouse AOM/DSS and DSS model of CAC.

The authors show that ITF2 and RELA/p65 expression are coordinatively upregulated in intestinal epithelial cells during acute colitis in mice, and that TNF α stimulation increases both ITF2 expression and NF- κ B activation in CRC cell lines (Fig. 1). Curiously, in CAC patients, ITF2 nuclear localization increased in dysplastic tissue relative to normal intestinal mucosa, but then decreased in carcinoma relative to dysplastic tissue. In contrast, RELA/p65 nuclear translocation steadily increased during the transition from normal to dysplastic tissue and further from dysplastic tissue to carcinoma (Fig. 7). The authors interpret these results as an indication that ITF2 plays a key role in the suppression of CAC progression in CD/UC patients by blocking NF- κ B activation.

These findings are of potential interest, and convincing evidence is presented that intestinal epithelial-specific ITF2 deletion promotes colonic inflammation and CAC progression in the mouse AOM/DSS model. However, several key points remain to be addressed.

1) Firstly, while ITF2 is clearly relevant to CAC inhibition in the AOM/DSS and DSS models, the clinical significance of the proposed mechanism in the control of CAC progression in CD/UC patients is not clear. It is also unclear from the limited nuclear localization data presented whether ITF2 is a key regulator of NF- κ B activity in these clinical settings and whether and how Parkin is involved in the regulation of ITF2 stability in human CD/UC. For instance, is there a correlation between ITF2 (or Parkin) expression and disease severity/clinical outcomes in UC/CD or CAC patients? Further, is there any evidence of ITF2 loss-of-function or Parkin gain-of-function mutations occurring in human CAC? Are there any SNPs in these genes or their regulators that might predispose UC/CD patients to CAC? Crucially, what downregulates ITF2 expression during the transition from dysplasia to carcinoma in CAC patients? Any data addressing at least some of these points would considerably strengthen the clinical relevance of the proposed mechanism.

2) Secondly, it is unclear that the observed effects of ITF2 in AOM/DSS-driven CAC are directly related to Parkin-dependent regulation or NF- κ B inhibition. a) The ITF2 interactions with Parkin and RELA/p65 were investigated in overexpression systems, and there is no evidence that endogenous proteins engage in these interactions either in *in vitro* cell culture systems or intestinal epithelial cells *in vivo*, which could be examined for instance by using PLA assays. b) Furthermore, the molecular interactions of ITF2 with RELA/p65 and Parkin are only superficially characterized, thus precluding the generation of mutant proteins with reduced binding affinity, which could provide useful tools to establish whether the biological effects of ITF2 on CAC suppression are indeed mediated by NF- κ B inhibition. Similarly, it would be helpful to determine whether Parkin conditional deletion in intestinal epithelial cells has the opposite phenotype of ITF2 deletion in AOM/DSS-dependent CAC and whether the effects of Parkin loss in this model are reversed by ITF2 deletion. Any data along these lines would significantly strengthen the manuscript conclusions and the proposed mechanism of action of ITF2 in suppressing CAC pathogenesis.

3) Thirdly, there are other technical limitations. For instance, many key experiments were conducted using a single CRC cell line, raising the possibility that the observed biological effects could have resulted from idiosyncrasies of that particular cell line. Moreover, many of the inhibitors used in Fig. 1 are not specific for their declared targets.

4) Lastly, the text presents some apparent inconsistencies and issues that have not been adequately discussed or explained. For example, it is stated that previous studies have shown

increased ITF2 mRNA expression in UC and CD patients relative to normal individuals, with a positive correlation with TNA levels (refs 22-24). Yet, the authors' own data show no changes in ITF2 mRNA levels between healthy individuals and UC/CD patients (Fig. 2I). Further, no changes in ITF2 mRNA expression were observed in CRC cell lines following TNF stimulation (Fig. 2A-2B).

Reviewer #2 (Remarks to the Author):

This is an interesting manuscript describing the role of Immunoglobulin transcription factor 2 (ITF2) in colonic inflammation and colitis associated tumorigenesis. ITF2, also known as TCF4, is a bHLH transcription factor that targets E boxes and prior studies suggest that it is a context dependent tumor suppressor or oncogene. ITF2 is upregulated in direct relation to TNF in IBD. p65 was also upregulated under inflammation and was demonstrated to stabilize ITF2 protein by disrupting Parkin (an E3 ubiquitin ligase):ITF2 interactions. The ITF2:p65 interaction domain mapped to N-terminus of ITF2 and carboxy tail of p65, inhibiting ubiquitination. They then demonstrate that epithelial deletion of ITF2 resulted in increased nuclear p65 and increased AOM/DSS induced tumorigenesis. ITF2, while upregulated in IBD, is lost in human CAC samples, whereas p65 was increased. Overall, these are well designed and executed experiments and an important discovery. I do have a few major and minor critiques.

Major critiques:

- 1) The microbial differences can have a large impact on DSS and AOM/DSS modeling. What measures were taken to control for microbial differences between experimental groups? These should be clearly stated in the methods.
- 2) The methods section indicates that half the colons were excised for RNA and protein analysis, and half for histopathologic analysis. DSS commonly has a larger impact on the distal colon, in fact that appears to be supported by looking at the tumor distribution in the AOM/DSS experiments (predominantly distal tumors). The authors should describe which "halves" were obtained. Proximal vs distal half or where the colon halved longitudinally? Also, if proximal and distal halves, how does this confound their work?
- 3) I am not sure that I completely follow the model picture. It seems that the last panel should be in the setting of cancer, not inflammation, demonstrating that loss of ITF2 results in increased nuclear p65 and upregulated target genes.
- 4) The authors convincingly demonstrate that ITF2 is upregulated with DSS treatment. Did they observe reductions with AOM/DSS in the tumors and what were its levels in the adjacent "normal". I would predict reduction in tumors coincident with p65 nuclear localization and presence in the adjacent normal. It would be interesting to know b/c if ITF2 was also lost in the "adjacent normal" it might suggest a field or priming effect with ITF2 loss.
- 5) Can the authors state if the organoid data is colon or small intestine. Treatment of colonoids is more relevant to the current report.

Minor:

- 1) Thorough proofing of the manuscript for typographical and grammar/usage errors should be performed.

Reviewer #3 (Remarks to the Author):

In this manuscript, Lee and coauthors identified that ITF2 (also known as TCF4) binds p65/RelA, a NF- κ B transcription factor, and that stabilizes ITF2. Furthermore, the authors showed that parkin is a ubiquitin ligase (E3) to induce proteasomal degradation of ITF2. Then, the authors concluded that the ITF2-p65-parkin axis plays an important role in colitis-associated cancer. Although interesting findings have been made in this study, the cellular and pathological mechanisms have not been fully elucidated.

Major comments:

1. In Fig. 1, the authors showed that ITF2 expression is positively correlated with p65/RelA levels.

Among five NF- κ B factors, RelB and c-Rel, but not p50 or p52, have similar domain structure with that of p65. Do RelB and c-Rel affect ITF2 binding, stabilization, and ubiquitination like p65? Please show the expression levels of p65 in Fig. 1C. In Fig. 1D, the pretreatment with TPCA1, an IKK inhibitor, seems to suppress TNF- α -induced IKK activation. Then, why are p65 and I κ B α were degraded without NF- κ B activation? In Fig. 1C and 1D, the bands visible on the border below the ITF2 blot appears to be augmented in the presence of Bay 11-7082 and TPCA1. It is necessary to submit uncropped immunoblotting images to clarify whether they are dephosphorylated ITF2 by the inhibition of IKK.

2. In Fig. 2E and 2F, the authors showed that cells treated with TNF exhibited a higher half-life of ITF2 than that without TNF- α . Upon TNF- α stimulation, p65 basically translocate into the nucleus and function as a transcription factor with shuttling to cytosol. Thus, it is unclear where the physiological p65-ITF2 binding occurs to stabilize ITF2. Moreover, in the presence of TNF- α and CHX, cells induce RIP1-independent apoptosis. Does ITF2 regulate TNF- α -mediated cell death pathways, such as apoptosis and necroptosis?

3. In Fig. 3, although the authors showed the interaction of p65 and ITF2 by overexpression system, the authors need to analyze the endogenous interaction of ITF2 and p65 with or without TNF- α stimulation.

4. In Fig. 4A and on lines 443-444, the authors described that parkin is the only E3 among ITF2-interacting proteins in the BioGRID software. However, when I checked the BioGRID database, TCF4 (ITF2) has 281 unique interactors (<https://thebiogrid.org/112787/summary/homo-sapiens/tcf4.html>), and besides PARK2, E3s such as CUL4B, IRF2BP2, RNF14, RNF6, TRIM24, TRIM68, UHRF2, and so on, are listed. Is parkin really the only E3 for ITF2? Basically, parkin is a latent E3, which is activated by PINK1-mediated phosphorylation of the parkin-UBL domain and/or ubiquitin. Is this kind of phosphorylation happening? Does mitochondrial damage that physiologically activates parkin affect the proteasomal degradation of ITF2? The authors need to investigate the parkin-induced ubiquitination site(s) in ITF2 and linkage of polyubiquitin chain. In vitro ubiquitination assay, as shown in Fig. 4D, is very weak. HeLa cells do not express parkin, but does ubiquitination occur when ITF2 and HA-Ub were co-expressed in HeLa cells? Moreover, to clarify the contribution of parkin in CRC, please show the expression levels of parkin in cells shown in Fig. S2A.

5. In Fig. 5C, the authors estimate the mechanism by which ITF2 suppresses the nuclear translocation of p65 by binding to p65, but this is an unstimulated state and cannot be evaluated. It is necessary to analyze the dynamics of p65 associated with inflammatory cytokine stimulation as mentioned above.

6. In Fig. 6, show p65 and parkin expression with IB and IHC. Differences in NF- κ B target genes expression and cell death induction in tumor and non-tumor areas also need to be analyzed.

Minor comments:

1. Please specify the TNF- α concentrations used for Fig. 1B and 2B.
2. Please specify cells used in Fig 4B.

Responses to the reviewers' comments regarding our manuscript entitled "Protein Stabilization of ITF2 by NF- κ B Prevents Colitis-associated Cancer Development"

(Manuscript ID: NCOMMS-21-47878)

Statement of Revision to Reviewer #1

We would like to begin by expressing our appreciation for the reviewer's thoughtful and considerate suggestions and comments. We hope our responses will be deemed satisfactory. Thank you for the reviewer's critical reading and suggestions for strengthening our conclusions.

Major comments

Question 1. Firstly, while ITF2 is clearly relevant to CAC inhibition in the AOM/DSS and DSS models, the clinical significance of the proposed mechanism in the control of CAC progression in CD/UC patients is not clear. It is also unclear from the limited nuclear localization data presented whether ITF2 is a key regulator of NF- κ B activity in these clinical settings and whether and how Parkin is involved in the regulation of ITF2 stability in human CD/UC. For instance, is there a correlation between ITF2 (or Parkin) expression and disease severity/clinical outcomes in UC/CD or CAC patients? Further, is there any evidence of ITF2 loss-of-function or Parkin gain-of-function mutations occurring in human CAC? Are there any SNPs in these genes or their regulators that might predispose UC/CD patients to CAC? Crucially, what downregulates ITF2 expression during the transition from dysplasia to carcinoma in CAC patients? Any data addressing at least some of these points would considerably strengthen the clinical relevance of the proposed mechanism.

Answer: We appreciate the reviewer for questioning the invaluable and fundamental issue. To investigate whether ITF2 is associated with the severity or clinical outcomes in ulcerative colitis (UC)/Crohn's disease (CD) or colitis-associated cancer (CAC) patients, we further obtained 15 human UC tissues (a new IRB approval number is included in the manuscript: 2105-055-1218) and then stained them against ITF2. As expected, ITF2 was merely observed

in the normal but highly increased in dysplastic regions (Supplementary Figure S15A). We then evaluated the disease severity based on ITF2 expression levels. The levels of ITF2 peaked in dysplastic regions of UC patients but were significantly downregulated in CAC patients. Most importantly, ITF2 expression was almost lost in the late stage (stages 3 and 4) of CAC patients (Supplementary Figure S15B), implying that ITF2 is correlated with the disease severity of CAC. When we compared the Parkin expression levels based on the Parkin IHC staining in CAC patient tissues, Parkin levels seem to be increased in the dysplastic/carcinoma regions in contrast with normal regions. However, there are no significant differences between dysplastic and carcinoma regions (Supplementary Figure S15C).

We included these results in Supplementary Figures S15A-C and revised the manuscript as follows;

Manuscript Page 24, Line 556,

..... We, thus, questioned whether ITF2 is associated with the disease severity in patients with UC and CAC. After obtaining additional 15 UC specimens, we stained ITF2 and then compared the disease severity based on ITF2 expression levels. Importantly, the patients in the late stage (stages 3 and 4) presented significant loss of ITF2 (Supplementary Figures S15A, B), suggesting that ITF2 is highly correlated with CAC disease severity. However, Parkin levels are comparable between dysplastic and carcinoma regions of the CAC patient tissues (Supplementary Figure S15C)...

In Supplementary Figure S15

Regarding the next question, although this issue is a crucial point of the manuscript and has to be further addressed, we cannot find appropriate publications about what the reviewer exactly mentioned. We do believe that there are, to our knowledge, any related papers showing evidence of ITF2 loss-of-function or regulators of ITF2. However, as shown in Figure 7 and Supplementary figure S15, we revealed that ITF2 protein expression levels are highly lost during the dysplastic-to-carcinoma transition. Especially, the patients who experienced severe CAC clinical symptoms (stages 3 and 4) presented much lower ITF2 levels compared to the early stages (stages 0-2). Consistently, the microarray dataset (GSE 3629, shown in Figure 7C) displayed rare ITF2 mRNA expressions in UC-associated neoplastic lesions, UC-associated carcinoma tissues, and sporadic CRC tissues, suggesting that the levels of ITF2 have already disappeared in the mRNA levels. We therefore highly suspect that ITF2 expression is genetically lost during the dysplasia-to-carcinoma transition in CAC patients as sporadic CRC patients did.^{1,2} For future works, we'll scrutinize whether ITF2 has LOF or GOF in CAC patients, and also it'll be interesting to find possible ITF2 regulators in the dysplastic-to-carcinoma transition. As the reviewer knows, it requires a huge time for recruiting appropriate patients (especially, CAC patients) and gathering related patient information. We understand

the reviewer's concerns but would ask that the reviewer show some leniency on this point. Because it is still an important issue, we later will do using a well-designed plan. We hope that this answer would satisfy your question.

References

1. Shin, H.W., Choi, H., So, D., Kim, Y.I., Cho, K., Chung, H.J., Lee, K.H., Chun, Y.S., Cho, C.H., Kang, G.H., Kim, W.H. & Park, J.W. ITF2 prevents activation of the beta-catenin-TCF4 complex in colon cancer cells and levels decrease with tumor progression. *Gastroenterology* **147**, 430-442 e438 (2014)
2. Herbst, A., Bommer, G.T., Kriegl, L., Jung, A., Behrens, A., Csanadi, E., Gerhard, M., Bolz, C., Riesenberger, R., Zimmermann, W., Dietmaier, W., Wolf, I., Brabletz, T., Goke, B. & Kolligs, F.T. ITF-2 is disrupted via allelic loss of chromosome 18q21, and ITF-2B expression is lost at the adenoma-carcinoma transition. *Gastroenterology* **137**, 639-648, 648 e631-639 (2009).

Question 2. Secondly, it is unclear that the observed effects of ITF2 in AOM/DSS-driven CAC are directly related to Parkin-dependent regulation or NF- κ B inhibition. a) The ITF2 interactions with Parkin and RELA/p65 were investigated in overexpression systems, and there is no evidence that endogenous proteins engage in these interactions either in in-vitro cell culture systems or intestinal epithelial cells in vivo, which could be examined for instance by using PLA assays. b) Furthermore, the molecular interactions of ITF2 with RELA/p65 and Parkin are only superficially characterized, thus precluding the generation of mutant proteins with reduced binding affinity, which could provide useful tools to establish whether the biological effects of ITF2 on CAC suppression are

indeed mediated by NF- κ B inhibition. Similarly, it would be helpful to determine whether Parkin conditional deletion in intestinal epithelial cells has the opposite phenotype of ITF2 deletion in AOM/DSS-dependent CAC and whether the effects of Parkin loss in this model are reversed by ITF2 deletion. Any data along these lines would significantly strengthen the manuscript conclusions and the proposed mechanism of action of ITF2 in suppressing CAC pathogenesis.

Answer: We appreciate the reviewer's thoughtful comment. Following the reviewer's suggestion, we investigated the ITF2 interactions with Parkin or p65 in the endogenous conditions and found similar results as we presented in the overexpression systems. These experiments were done in two different CRC cell lines (colo320DM and SW480) (Supplementary Figures S10A, B). Besides, the endogenous interaction between ITF2 and p65 was notably increased following the TNF- α treatment (Supplementary Figure S10D). We also exploited the PLA assay as the reviewer recommended to examine whether those proteins bind to each other, and observed similar results (Supplementary Figure S10C). We included these results in supplementary Figures S10A-C and revised the manuscript as follows;

Manuscript Page 13, Line 291,

... ***In Vitro* PLA assay**

Duolink fluorescent reagents were purchased from Sigma-Aldrich (MO, USA) and performed as specified by the manufacturer...

Manuscript Page 21, Line 493,

...Consistently, co-immunoprecipitation assay against endogenous proteins as well as PLA assay indicated that ITF2 interacts with p65 and Parkin (Supplementary Figures S10A-C). The

endogenous interaction between p65 and ITF2 was notably increased following the TNF stimulation (Supplementary Figure S10D)...

In Supplementary Figure S10

Lee et al, Fig. S10

Regarding the second question, we decided to identify the Parkin binding sites to ITF2 to see the molecular interactions deeply. We carried out the in vitro binding assay and confirmed that ITF2 and Parkin bind together (Figure 4C). Immunoprecipitation assay showed that Parkin binds to the N-terminals of the ITF2 (Figure 4D). We then utilized one of the 3D computational protein binding prediction programs (AlphaFold 2) to characterize promising amino acids to which Parkin can bind to the ITF2. AlphaFold2 results revealed that Parkin binds to the N-terminals of the ITF2 between the lysine residues (171-175) (Figure 4E). After generating the

ITF2 mutant which is deleted in 5-amino acids (Lys171-175), we then transfected the mutant ITF2 into the HeLa cells which have been reported to be little or no endogenous Parkin expression^{1,2} in the absence or presence of Parkin to determine whether these lysine residues are the major ubiquitination sites of the ITF2 by the Parkin. As expected, Parkin induced ITF2 ubiquitination, but the levels seem to be downregulated when the cells are transfected with ITF2 mutant (Δ K171-175), suggesting that the lysine residues between the 171-175 of the ITF2 are the major ubiquitination sites by the Parkin (Figure 4F and Supplementary Figure S9D). We included those results in Figure 4 and Supplementary Figure S9, and modified the manuscript as follows;

Manuscript Page 9, Line 199,

...Protein-protein binding sites prediction

Alphafold2 predicts three-dimensional (3D) protein structures based on the protein sequences of the ITF2 and Parkin, and then showed promising Lys residues that are supposed to be ubiquitinated by the Parkin in the N-terminus of the ITF2.³⁰

Manuscript Page 15, Line 342,

... Site-specific deletion mutation of ITF2 was performed using PCR-based mutagenesis...

Manuscript Page 20, Line 470,

...Among the 284 unique ITF2-interacting proteins displayed in the BioGRID, we found 9 candidates including PARK2 which have an E3 ubiquitin-protein ligase role (Figure 4A), and then we narrowed down to the 3 promising targets based on the experimental evidence which was demonstrated by the two-hybrid system. However, the knockdown experiment presented that only Parkin was responsible for the ITF2 ubiquitination (Supplementary Figures S9A-C).

Reversely, overexpression of Parkin with ITF2 in the HeLa cells which have little or no endogenous Parkin expression,^{39,40} induced ITF2 ubiquitination (Supplementary Figure S9D). Co-immunoprecipitation and in vitro binding assay confirmed that Parkin can bind to ITF2 (Figures 4B, C), and it turns out to be that Parkin binds to the N-terminals of ITF2B (Figure 4D). To corroborate which sites of ITF2 are associated with Parkin-induced ITF2 ubiquitination, we generated an ITF2 mutant where key lysine residues (Lys171-175) are removed based on the AlphaFold2 results (Figure 4E). As expected, when the HeLa cells are overexpressed with Parkin and WT ITF2, we observed upregulation of ITF2 ubiquitination. However, transfection with ITF2 mutant and Parkin revealed attenuated ubiquitination capacity (Figure 4F).....

In Figure 4

For the final question, as the reviewer knows, Parkin has been reported to activate NF- κ B activity in multiple different tissues regardless of ITF2 expression^{3,4,5}, so we could expect reduced inflammation, diminished tumor numbers, or possibly, decreased tumor sizes in the Parkin cKO mice. However, even though Parkin cKO mice showed the opposite results compared to the results presented in ITF2 cKO mice, we might think that it is hard to say Parkin

deletion reduced NF- κ B activity through the ITF2 stabilization method. As we discussed in question 1, we introduced Parkin as a novel E3 ligase for ITF2 ubiquitination, but we highly suspect that ITF2 seems to be genetically lost during dysplasia-to-carcinoma transition rather than Parkin-dependent mechanisms (Figure 7). In addition, when we take a look at the Parkin expression patterns in the CAC patient tissues, there is no significant difference between dysplasia and carcinoma regions (Supplementary Figure S15C). We understand the reviewer's concern and it would have been better if we could generate double knock-out mice (e.g. generate Parkin cKO in the ITF2cKO mice) and check whether those results could happen. Indeed, we were looking into the Parkin floxed mice to generate intestinal-specific parkin deletion mice but, unfortunately, it was not commercially available and the timeline was not allowed us to get or generate a new mouse strain. Besides, the COVID period makes all things much slower. Hopefully, this answer would satisfy your question.

References

1. Denison SR, *et al.* Alterations in the common fragile site gene Parkin in ovarian and other cancers. *Oncogene* **22**, 8370-8378 (2003).
2. Pawlyk AC, *et al.* Novel monoclonal antibodies demonstrate biochemical variation of brain parkin with age. *J Biol Chem* **278**, 48120-48128 (2003).
3. Wang Y, Shan B, Liang Y, Wei H, Yuan J. Parkin regulates NF- κ B by mediating site-specific ubiquitination of RIPK1. *Cell Death Dis* **9**, 732 (2018).
4. Henn IH, *et al.* Parkin mediates neuroprotection through activation of I κ B kinase/nuclear factor- κ B signaling. *J Neurosci* **27**, 1868-1878 (2007).
5. Sha D, Chin LS, Li L. Phosphorylation of parkin by Parkinson disease-linked kinase PINK1 activates parkin E3 ligase function and NF- κ B signaling. *Hum Mol Genet* **19**, 352-363 (2010).

Question 3. Thirdly, there are other technical limitations. For instance, many key experiments were conducted using a single CRC cell line, raising the possibility that the observed biological effects could have resulted from idiosyncrasies of that particular cell line. Moreover, many of the inhibitors used in Fig. 1 are not specific for their declared targets.

Answer: We appreciate the reviewer's thoughtful comments which can improve our manuscript. Following the reviewer's suggestion, we carefully go over the key experiments which were performed in a single colorectal cancer (CRC) cell line and noticed that Figures 1C, and D were conducted in colo320DM cells only. Except for these, we might think that all of the experiments were done on at least two different CRC cell lines or carried out in an overexpression system to reveal underlying mechanisms. We agree with the reviewer's opinion that many of the inhibitors described in Figure 1C are not that specific for their declared target but rather affect multiple targets. For example, Rapamycin majorly inhibits the mTOR signaling pathway but also could affect multiple targets which were responsible for cell growth and cell cycle progression. In the same context, AG490 partially affects the NF- κ B pathway as well as blocks the JAK-STAT pathway. However, those inhibitors mentioned in the manuscript are highly selective for the described target (the declared target means major target, not means single target), thus various researchers used those inhibitors for the same purposes we do believe. Most importantly, the NF- κ B inhibitor (Bay 11-7082) shown in Figure 1C was the only chemical that could attenuate TNF-induced ITF2 upregulation even though the other inhibitors might affect multiple signaling pathways. As mentioned above, AG490 reportedly could partially inhibit the NF- κ B activation and that is why AG490 pre-incubation presented a half reduction of the TNF-induced ITF2 expressions. To reduce the reviewer's concern that the

observed biological effects could have resulted from idiosyncrasies of that particular cell line, we looked into the other kinds of NF- κ B inhibitors such as TPCA1 and Bay-11-7085 (reportedly, much more specific for NF- κ B) and then performed similar experiments in two additional CRC cell lines (SW480 and WiDr) including Colo320DM cells (Figure 1D and Supplementary Figure S3). TPCA1 and Bay-11-7085 have been known to be potent IKK inhibitors that enable to block of TNF- α -induced phosphorylation of I κ B- α , resulting in an inhibition of NF- κ B. As expected, pre-incubation of the TPCA1 or Bay-11-7085 lessened TNF-induced ITF2 upregulation as Bay-11-7082 did, suggesting that NF- κ B is involved in the TNF-induced ITF2 expression. In addition, p65 knockdown and overexpression studies presented in Figure 1-4 confirmed that p65 is responsible for TNF-induced ITF2 expression. As reviewer 3 requested p65 immunoblot and uncropped immunoblot of ITF2 in Figure 1C (Question 1) to examine whether the bands visible on the border below of the ITF2 blot related to dephosphorylation issue, we reconducted this experiment using the same samples to check whether this issue is a technical problem or dephosphorylation. To avoid any confusion, we modified the previous figures to the current ones.

We added these data to Figures 1C, D, and Supplementary Figure S3, and modified the manuscript as follows;

Manuscript Page 10, Line 228,

... TPCA1 (20 μ M), BAY 11-7085 (10 μ M) ...

Manuscript Page 17, Line 390,

... We then investigated which signaling pathways are associated with the TNF-mediated ITF2 expression by using representative known inhibitors such as mTOR (rapamycin), MEK/ERK (PD98059), AKT/p38 (SB203580), JAK-STAT (AG490), AKT (MK2206) and NF- κ B (Bay11-

7082). The NF-κB inhibitor (Bay11-7082) significantly blocked TNF-induced ITF2 expression (Figure 1C). For confirmation, we additionally added inhibitors to block NF-κB (Bay11-7085, TPCA1) and then examined whether this finding is repeatedly observed in different colon cancer cell lines. SW480 and WiDr cell lines including Colo320DM which can express ITF2 showed a similar result (Figure 1D and Supplementary Figure S3)...

In Figure 1,

In Supplementary Figure S3

Question 4. Lastly, the text presents some apparent inconsistencies and issues that have not been adequately discussed or explained. For example, it is stated that previous studies have shown increased ITF2 mRNA expression in UC and CD patients relative to normal

individuals, with a positive correlation with TNA levels (refs 22-24). Yet, the authors' own data show no changes in ITF2 mRNA levels between healthy individuals and UC/CD patients (Fig. 2I). Further, no changes in ITF2 mRNA expression were observed in CRC cell lines following TNF stimulation (Fig. 2A-2B).

Answer: We apologize for this confusion and we agree that these inconsistencies have to be addressed in the discussion part. As the reviewer's pointed out, previous reports (refs 22-24) showed increased ITF2 mRNA expression in both UC and CD patients compared to healthy individuals whereas our data (Fig. 2A, B) and the data from GEO databases (Fig. 2I) revealed that the levels of ITF2 are comparable across the samples.

If we take a guess, we might think that the reason is coming from the inconsistent sample collection method from the patient biopsies. The reference papers (refs 22-24) and most of the GEO datasets mentioned that they used biopsies from the colonic mucosa derived from healthy, UC, or CD which enables possible contamination of lymphocytes. It is unclear how well the sample collection was controlled. For example, some of the UC or CD samples were taken from relatively normal regions but sometimes, the samples could contain relatively high inflammatory locations. In addition, there is a possibility that the authors did epithelial cell sorting before running the bulk RNA-seq. As the reviewer knows, ITF2 reportedly binds to the immunoglobulin enhancer Mu-E5/KE5-motif to facilitate immunoglobulin expression. Thus, if the RNAs were extracted from the colonic mucosa, and not even sorted, mucosa samples could randomly contain lymphocytes, making it hard to interpret the data. That is why many of the sequencing data showed contradictory results. However, most importantly, our in-vitro data (Figure 2A, B) and GSE 11223 dataset which was done in epithelial biopsy showed that the ITF2 mRNA levels were comparable across the groups.

We discussed this issue in the discussion part as follows;

...There is an inconsistency that whether the mRNA levels of ITF2 are upregulated in UC/CD patients. Some of the previous reports performed microarray analysis using colon mucosal biopsies and showed increased ITF2 mRNA levels in UC and CD patients relative to a healthy individuals.^{23, 24, 46} In addition, Noble et al presented upregulated ITF2 mRNA expressions with a positive correlation with TNF levels in UC tissues.²² On the other hand, Figures 2A, B and many of the other GEO datasets in Figure 2I exhibited no significant changes in ITF2 mRNA levels across the samples. We might suspect that this conflict result would be coming from the inconsistent sample collection method from the patient tissues. The papers mentioned above mostly utilized biopsies from the colonic mucosa derived from healthy, UC, or CD patients which enable possible contamination of lymphocytes. Besides, there is a chance that samples might be collected in the randomly selected areas, which means the ratio of the lymphocytes and epithelial compartment could be arbitrarily decided. Thus, if the RNAs were extracted from the colonic mucosa, and not even sorted, mucosa samples could randomly contain lymphocytes, making it hard to interpret the data. That is why many of the sequencing data showed contradictory results. However, most importantly, our in-vitro data (Figure 2A, B) and GSE 11223 dataset which was done in epithelial biopsies showed that the ITF2 mRNA levels were comparable....

Statement of Revision to Reviewer #2

We authors thank the reviewer for the comments, which contributed to the improvement of this manuscript.

Major comments

Question 1. The microbial differences can have a large impact on DSS and AOM/DSS modeling. What measures were taken to control for microbial differences between experimental groups? These should be clearly stated in the methods.

Answer: We agree with your opinion that microbial differences could highly impact DSS and AOM/DSS mouse models. One report showed that Germ-Free (GF) mice developed significantly more and larger tumors compared with that Specific-Pathogen-Free (SPF) mice after AOM and DSS treatment.¹ In addition, two different papers also presented that both IL-2-deficient and IL-10-deficient mice, which under conventional conditions develop spontaneous colitis, have significantly reduced or absent intestinal inflammation in germ-free conditions.^{2,3} To sustain a similar condition, all the mice treated with DSS and/or AOM/DSS were bred and maintained in SPF conditions. SPF mice were fed the same autoclaved chow diet and used the same drinking water offered by the SPF room. As all animal facilities do, our animal facility routinely (mostly at 6w intervals) runs PCR, ELISA, or culture to monitor possible bacterial & mycoplasma contamination. Therefore, we do believe that our experiments were done at least under similar conditions. We already described some information in the method section of the main manuscript, but we also included this information as follows;

... All mice used in this study were housed at the specific-pathogen-free (SPF) animal facilities with a controlled temperature and free access to food and water under a 12-hour light: dark cycle. To avoid bacteria infection & mycoplasma contamination, and maintained the mouse condition similarly as much as we can, we used the autoclaved chow diet and drinking water offered by the animal facility....

References.

1. Zhan Y, *et al.* Gut microbiota protects against gastrointestinal tumorigenesis caused by epithelial injury. *Cancer Res* **73**, 7199-7210 (2013).
2. Sellon, R. K. *et al.* Resident enteric bacteria are necessary for development of spontaneous colitis and immune system activation in interleukin-10-deficient mice. *Infect Immun* **66**, 5224-5231, doi:10.1128/IAI.66.11.5224-5231.1998 (1998).
3. Schultz, M. *et al.* IL-2-deficient mice raised under germfree conditions develop delayed mild focal intestinal inflammation. *Am J Physiol* **276**, G1461-1472, doi:10.1152/ajpgi.1999.276.6.G1461 (1999).

Question 2. The methods section indicates that half the colons were excised for RNA and protein analysis, and a half for histopathologic analysis. DSS commonly has a larger impact on the distal colon, in fact that appears to be supported by looking at the tumor distribution in the AOM/DSS experiments (predominantly distal tumors). The authors should describe which “halves” were obtained. Proximal vs distal half or where the colon halved longitudinally? Also, if proximal and distal half, how does this confound their work?

Answer: We are sorry about the confusion. As mentioned in the manuscript, we excised the colon horizontally (proximal vs distal), and then took the distal colons for histology (Figures 1, 6 and Supplementary Figures S4, S11, S12, and S13) and used the other half for protein work (western blot, only for Figure 6B). Figure 6B was presented to show whether ITF2 is deleted in the colonic epithelial cells specific manner. We, therefore, might think that the tissues coming from the other half (proximal part) were not going to be a problem as ITF2 knockdown was evaluated in the same proximal part of the tissues. We also took some portion of the proximal tissues and then extracted the RNA, but have not used it in all of our experiments. We just stored the RNA samples just in case. However, based on the reviewer's opinion, we should have divided the tissues as longitudinally. We'll reference it for future work. To avoid any confusion, we deleted the information on how we can get RNA from the colon as follows;

Manuscript Page 9, Line 190

... The half of the colon (proximal part) was taken as a tissue sample for subsequent protein analysis by isolating crypt epithelial cells from tissues. The other half (distal colon) was fixed with 4% PFA (Biosesang, Seongnam, South Korea) for paraffin embedding and histological analysis.

Question 3. I am not sure that I completely follow the model picture. It seems that the last panel should be in the setting of cancer, not inflammation, demonstrating that loss of ITF2 results in increased nuclear p65 and upregulated target genes.

Answer: We appreciate your kind opinion. Based on the reviewer's suggestion, we changed the wording "inflammatory condition" to "colitis-associated cancers (CAC) with ITF2 (-).

We modified the graphical summary as follows;

In Figure 7D,

Editorial Note: Figure below created with BioRender.com.

D

Question 4. The authors convincingly demonstrate that ITF2 is upregulated with DSS treatment. Did they observe reductions with AOM/DSS in the tumors and what were its levels in the adjacent “normal”. I would predict the reduction in tumors coincident with p65 nuclear localization and presence in the adjacent normal. It would be interesting to know b/c if ITF2 was also lost in the “adjacent normal” it might suggest a field or priming effect with ITF2 loss.

Answer: We appreciate the invaluable comments. Following the reviewer's suggestion, we looked into ITF2 expression levels in the AOM/DSS-treated WT mice and evaluated whether ITF2 expressions would be changed in adjacent normal regions. As expected, the adjacent normal areas presented upregulated ITF2 expressions (AOM/DSS-treated mice). However, when we move the areas closer to the tumor areas, the levels of ITF2 seem to be downregulated and are highly lost in tumor regions (See below). Indeed, these patterns are quite similar as shown in human colitis-associated-cancer (CAC) tissues (Figure 7A).

Question 5. Can the authors state if the organoid data is colon or small intestine. Treatment of colonoids is more relevant to the current report.

Answer: We apologize for providing insufficient information. We performed the organoid culture using crypt epithelial cells from the mouse colon tissues. Based on the reviewer’s suggestion, we changed the wording “IEC organoids” to “colonic organoids” for making sure where they were coming from.

We revised the manuscript as follows;

Manuscript Page 7, Line 160,

.....For the acute colitis model and **colonic organoids.....**

Manuscript Page 10, Line 225,

Colonic organoids...

Manuscript Page 12, Line 272,

IEC isolation and culture of colonic organoids....

Manuscript Page 12, Line 273,

Mouse colonic epithelial cells were isolated.....

Main Figure 1B legend,

western blot analysis of ITF2 in the colonic organoids treated with TNF (20 ng/ml) in a concentration-dependent

Minor comments

1) Thorough proofing of the manuscript for typographical and grammar/usage errors should be performed.

Answer: Based on the reviewer's suggestion, we revised the manuscript to change typographical and grammatical errors.

Statement of Revision to Reviewer #3

We thank the reviewer for raising these issues and hope that our response will be satisfactory.

Major comments

Question 1. In Fig. 1, the authors showed that ITF2 expression is positively correlated with p65/RelA levels. Among five NF- κ B factors, RelB and c-Rel, but not p50 or p52, have a similar domain structure to that of p65. Do RelB and c-Rel affect ITF2 binding, stabilization, and ubiquitination like p65? Please show the expression levels of p65 in Fig. 1C. In Fig. 1D, the pretreatment with TPCA1, an IKK inhibitor, seems to suppress TNF- α -induced IKK activation. Then, why are p65 and I κ B α degraded without NF- κ B activation? In Fig. 1C and 1D, the bands visible on the border below the ITF2 blot appear to be augmented in the presence of Bay 11-7082 and TPCA1. It is necessary to submit uncropped immunoblotting images to clarify whether they are dephosphorylated ITF2 by the inhibition of IKK.

Answer: We are thankful for the reviewer's kind comments. Following the reviewer's suggestion, we first examined whether RelB and c-Rel could bind to the ITF2, and then evaluated whether these proteins were also associated with the stabilization and ubiquitination of ITF2 as p65 did. Although RelB and c-Rel can bind to the ITF2 (Supplementary Figure S8A), they cannot affect the expression and ubiquitination levels of the ITF2 (Supplementary Figures S8B, C). Taken together, among the three different proteins that have similar domain structures, we suggest that p65 is the only protein that can affect ITF2 expression levels by regulating ubiquitination.

We included those results in Supplementary Figure S8, and modified the manuscript as follows;

... As RelB and c-Rel reportedly, not p50 and p52, have similar domain structures to that of p65 among five NF- κ B factors, we furtherly evaluated whether RelB and c-Rel are associated with the stabilization and ubiquitination of ITF2 as p65 did. Even though RelB or c-Rel bind to the ITF2, they cannot affect the expression and ubiquitination levels of the ITF2 (Supplementary Figures S8A-C)....

Manuscript Page 15, Line 341,

..... RelB and C-Rel plasmids were purchased from OriGene (Rockville, MD, USA).....

In Supplementary Figure S8,

Lee et al, Fig. S8

We also included the p65 immunoblot in Figure 1C as the reviewer requested. As shown in Figure 1D, when the cells were pretreated with NF-κB inhibitor (Bay 11-7082), the p65 expression level was downregulated. To confirm this issue, we included additional NF-κB inhibitors such as TPCA1, Bay 11-7085, and then examined those effects on p65 in three different CRC cell lines (Colo320, SW 480, WiDr). We observed very similar results in each cell line (Figure 1D, Supplementary Figure S3)

We included these figures and modified the manuscript as follows;

In Figures 1C, D

In Supplementary Figure S3,

Manuscript Page 10, Line 228,

.....MK2206 (5 μ M), BAY 11-7082 (10 μ M), TPCA1 (20 μ M), BAY 11-7085 (10 μ M) or MG132 (10 μ M) for 1 hour followed by TNF (20 ng/mL) stimulation....

Regarding the next question, p65 and I κ B- α seem to be degraded by the BAY 11-7082 and Bay 11-7085 as well as TPCA1 even though the samples were treated with TNF (Figures 1 C, D, and Supplementary Figure S3). We truly have zero clues why those inhibitors alleviate p65 and I κ B- α expression levels, but when we confirmed these issues in three different CRC cell lines, we observed similar phenomena as we showed before (Figure 1D and Supplementary Figure S3). Although we cannot explain the underlying mechanisms of how those inhibitors affect p65 and I κ B- α reductions, previous publications also showed I κ B- α reduction by the TPCA1 or Bay 11-7082 incubation^{1,2}. However, p65 seems to be quite stable even after the treatment of inhibitors. We speculate that these might be distinct features in the colonic cells but not occurred in other cell types or there is a chance that p65 has a positive feedback loop to maintain the p65 expression itself. Thus, interrupting p65 activation by the inhibitors might affect p65 expression itself. Hopefully, this answer and our additional experiment would satisfy your question.

For the last question, we re-electrophoresed the samples shown in Figure 1C to resolve whether the bands which were visible on the border below the ITF2 blot in Figure 1C are the dephosphorylated forms of ITF2 or technical errors. So, we decided to elongate the electrophoresed time around 1 h to make more room between the ITF2 band and the band visible on the border below the ITF2 as the two different bands were too close to evaluate something. However, when we ran the samples for a much longer time, “the expected dephosphorylated forms of ITF2” go far down more than 20kDa, and that results led us to think

about the technical errors. Besides, NF- κ B inhibitors used in this study have been known to be representative specific inhibitors for blocking I κ B- α phosphorylation by inhibiting IKK. As the reviewer knows, IKK is tightly regulated, highly stimulus-specific, and target-specific (which means I κ B- α) which is essential for a plethora of functions attributed to NF- κ B. To avoid any confusion, we changed the current data to a newly modified one which was electrophoresed longer time.

We added those results in Figures 1C, D, and Supplementary Figure S3. In addition, we included uncropped images in Supplementary Figure S16.

Supplementary Manuscript Page 31, Line 357

...Figure S16. Original images of western blots in the study. Uncropped blots represented with protein names and molecular weight markers...

In Supplementary Figure S16,

Figure S3C

References

1. Wang B, *et al.* TPCA-1 negatively regulates inflammation mediated by NF-kappaB pathway in mouse chronic periodontitis model. *Mol Oral Microbiol* **36**, 192-201 (2021).
2. Rauert-Wunderlich H, *et al.* The IKK inhibitor Bay 11-7082 induces cell death independent from inhibition of activation of NFkappaB transcription factors. *PLoS One* **8**, e59292 (2013).

Question 2. In Fig. 2E and 2F, the authors showed that cells treated with TNF exhibited a higher half-life of ITF2 than that without TNF- α . Upon TNF- α stimulation, p65

basically translocates into the nucleus and function as a transcription factor with shuttling to cytosol. Thus, it is unclear where the physiological p65-ITF2 binding occurs to stabilize ITF2. Moreover, in the presence of TNF- α and CHX, cells induce RIP1-independent apoptosis. Does ITF2 regulate TNF- α -mediated cell death pathways, such as apoptosis and necroptosis?

Answer: We might think that questions 2 and 5 have been made in the same context. To resolve the first question, we investigated whether p65 would be still left in the cytosol even after the TNF challenge and could have a chance to bind with ITF2. We simply stimulated the HCT116 cells with TNF and then performed a nuclear fraction experiment. If we take a look at lanes 1 and 2 in both cytosol and nuclear fraction, still plenty of p65 exists in the cytosol and they might have a chance for physiological binding with ITF2 (Figure 5D). Most importantly, when we conducted a similar experiment by using ITF2-overexpressed stable cells (HCT116) to evaluate the role of ITF2 in inflammatory conditions, the translocated p65 amount to the nucleus by the TNF treatment was downregulated by the ITF2 overexpression (Figure 5D), implying that ITF2-p65 complex in the cytosol interrupts p65 activation under inflammatory conditions.

We added this data to Figure 5D and revised the manuscript as follows;

Manuscript Page 11, Line 241,

.....Fractionation of cytoplasmic and nuclear components

HCT 116 stable cell lines established with small hairpin RNA targeting ITF2²⁰ were used and treated with or without TNF (Peprotech, Rocky Hill, CT, USA). The nuclear fraction experiment was performed as previously described.³²

Manuscript Page 22, Line 514,

...We also conducted a similar experiment under TNF-treated conditions to mimic inflammatory conditions. We observed that plenty of p65 was left in the cytosol even after TNF treatment, implying that p65 has a chance for physiological binding with ITF2 in the cytosol under inflammatory conditions. Most importantly, the amount of translocated p65 to the nucleus by the TNF challenge was downregulated by the ITF2 overexpression (Figure 5D)...

In Figure 5D,

Regarding the second question, we transfected the Colo320DM cells with ITF2 followed by the TNF stimulation for 8h or 24h respectively. Annexin V-PI staining results revealed that the proportion of the apoptotic cells and necrotic cells was comparable in the early timepoint (8 h), but apoptotic cells were reduced at a half ratio when they were incubated with TNF for 24 h. As the necrotic cell population seems to be comparable, we didn't measure the necroptosis cell population in the same condition. We might conjecture that 8h stimulation of TNF is too short to make any changes (the timepoint used in our study).

We included this data in Supplementary Figure S5 and revised the manuscript as follows;

Supplementary Manuscript Page 5, Line 116,

...FACS analysis of cell death

The effect of ITF2 on TNF-mediated cell death pathways such as apoptosis and necrosis was examined by the combined application of Annexin V-fluorescein isothiocyanate (FITC) apoptosis detection kits (BD Pharmingen, Heidelberg, Germany) and propidium iodide (Sigma-Aldrich, MO, USA) followed by flow cytometric analysis according to the manufacturer's instructions. Briefly, the Colo320DM cells were transfected with Myc-ITF2 followed by the TNF stimulation for 8 h, and 24 h respectively. The cells were resuspended in a binding buffer and then stained with FITC-conjugated annexin V antibody for 15 minutes in the dark at room temperature, and then incubated with propidium iodide sequentially. Flow cytometric analysis was immediately performed in a BD FACS Canto Flow Cytometer (BD Biosciences, San Jose, CA)....

Main manuscript Page 18, Line 422,

...As TNF and CHX incubation could induce RIP1-independent apoptosis in the cells, we simply examined whether ITF2 affects TNF-induced cell death pathways. Annexin V-PI staining results presented that there are any significant changes across the samples in the early time point, but the proportion of apoptotic cells was highly reduced in 24 h stimulatory condition. (Supplementary Figure S5)....

In Supplementary Figure S5

Question 3. In Fig. 3, although the authors showed the interaction of p65 and ITF2 by overexpression system, the authors need to analyze the endogenous interaction of ITF2 and p65 with or without TNF- α stimulation.

Answer: As the reviewer commented, we performed an immunoprecipitation analysis to see the interaction of p65 and ITF2 in endogenous conditions with or without TNF stimulation. Consistently, they were bound together and the binding amount was increased by the TNF challenge.

We added these results in Supplementary Figure S10, and modified the manuscript as follows;

Manuscript Page 21, Line 493,

...Consistently, co-immunoprecipitation assay against endogenous proteins as well as PLA assay indicated that ITF2 interacts with p65 and Parkin (Supplementary Figures S10A-C). The

endogenous interaction between p65 and ITF2 was notably increased following the TNF stimulation (Supplementary Figure S10D)....

In Supplementary Figure S10,

Lee et al, Fig. S10

Question 4. In Fig. 4A and on lines 443-444, the authors described that parkin is the only E3 among ITF2-interacting proteins in the BioGIRD software. However, when I checked the BioGIRD database, TCF4 (ITF2) has 281 unique interactors (), and besides PARK2, E3s such as CUL4B, IRF2BP2, RNF14, RNF6, TRIM24, TRIM68, UHRF2, and so on, are listed. Is parkin really the only E3 for ITF2? Basically, parkin is a latent E3, which is

activated by PINK1-mediated phosphorylation of the parkin-UBL domain and/or ubiquitin. Is this kind of phosphorylation happening? Does mitochondrial damage that physiologically activates parkin affect the proteasomal degradation of ITF2? The authors need to investigate the parkin-induced ubiquitination site(s) in ITF2 and the linkage of the polyubiquitin chain. In vitro ubiquitination assay, as shown in Fig. 4D, is very weak. HeLa cells do not express parkin, but does ubiquitination occur when ITF2 and HA-Ub were co-expressed in HeLa cells? Moreover, to clarify the contribution of parkin in CRC, please show the expression levels of parkin in cells shown in Fig. S2A.

Answer: We appreciate the reviewer for the important comment. Among the E3s that the reviewer mentioned, we firstly screened the E3 proteins based on the experimental evidence which was done by the two-hybrid system rather than the other methods, and then we finally got 3 proteins (PARK2, RNF6, and TRIM68) (Figure 4A). However, PARK2 was the only E3s that are responsible for the ubiquitination of the ITF2 as the knockdown of the RNF6 or TRIM68 cannot affect ITF2 ubiquitination (Supplementary Figures S9A-C, we included qPCR data instead of western blot to check knockdown efficiency as the antibody quality for RNF6 and Trim68 was too low). Consistently, when we overexpressed the PARK2 with ITF2 in HeLa cells which have been known to be little or no endogenous Parkin expression^{1,2} the levels of ITF2 ubiquitination were markedly upregulated (Supplementary Figure S9D).

To determine which sites are crucial for Parkin-mediated ITF2 ubiquitylation, we conducted an immunoprecipitation experiment and observed that Parkin binds to the N-terminals of the ITF2 (Figures 4C, D). We then exploited one of the 3D computational protein binding prediction programs (AlphaFold2) to identify promising lysine residues to which parkin could bind to the ITF2. AlphaFold2 results presented that the lysine residues (Lys171-175) of the ITF2 seem to be promising (Figure 4E). We, therefore, generated the ITF2 mutant (Δ K171-175) and then transfected it into the HeLa cells in the absence or presence of Parkin.

Importantly, the levels of ubiquitination by the Parkin are downregulated when the cells are transfected with mutant ITF2, suggesting that the lysine residues between the 171-175 of the ITF2 are the major ubiquitination sites by the Parkin (Figure 4F). As we investigated the parkin-induced ubiquitination sites in ITF2 which is more important than the in-vitro binding of ITF2 and Parkin, we moved in-vitro binding data to Supplementary Figure S9E.

Based on the experimental results, we added those results in Figure 4 and Supplementary Figure S9, and revised the manuscript as follows;

Manuscript Page 9, Line 199

..Protein-protein binding sites prediction

Alphafold2 predicts three-dimensional (3D) protein structures based on the protein sequences of the ITF2 and Parkin, and then showed promising Lys residues that are supposed to be ubiquitinated by the Parkin in the N-terminus of the ITF2.³⁰

Manuscript Page 15, Line 342

..Site-specific deletion mutation of ITF2 was performed using PCR-based mutagenesis...

Manuscript Page 20, Line 470

...Among the 284 unique ITF2-interacting proteins displayed in the BioGRID, we found 9 candidates including PARK2 which have an E3 ubiquitin-protein ligase role (Figure 4A), and then we narrowed down to the 3 promising targets based on the experimental evidence which was demonstrated by the two-hybrid system. However, the knockdown experiment presented that only Parkin was responsible for the ITF2 ubiquitination (Supplementary Figures S9A-C). Reversely, overexpression of Parkin with ITF2 in the HeLa cells which have little or no endogenous Parkin expression,^{39, 40} induced ITF2 ubiquitination (Supplementary Figure S9D).

Co-immunoprecipitation and in vitro binding assay confirmed that Parkin can bind to ITF2 (Figures 4B, C), and it turns out to be that Parkin binds to the N-terminals of ITF2B (Figure 4D). To corroborate which sites of ITF2 are associated with Parkin-induced ITF2 ubiquitination, we generated an ITF2 mutant where key lysine residues (Lys171-175) are removed based on the AlphaFold2 results (Figure 4E). As expected, when the HeLa cells are overexpressed with Parkin and WT ITF2, we observed upregulation of ITF2 ubiquitination. However, transfection with ITF2 mutant and Parkin revealed attenuated ubiquitination capacity (Figure 4F)...

In Figure 4,

In Supplementary Figure S9,

We added the Parkin immunoblot in Supplementary Figure S2A following the reviewer's suggestion.

In Supplementary Figure S2A

A
Lastly, as the reviewer mentioned, Parkin has known to be activated by the Pink1 which has unique features that enable it to phosphorylate ubiquitin and the ubiquitin-like (UBL) domain of Parkin (Ser 65), and the Pink1 is reportedly activated by mitochondrial membrane potential depolarization.³ It is unclear whether mitochondrial damage is directly associated with ITF2 loss in CAC patients. However, it is not surprising that there is a strong correlation between mitochondria dysfunction and cancer cell growth or tumorigenesis as cancers alter mitochondria functions to adapt to the surrounding environment.⁴⁻⁶ In addition, the previous paper shows the role of mitochondria defects in IBD and colorectal cancer.⁷ Taken together, mitochondria damage by the various kinds of stimuli in the colon cancer environment might modify the mitochondria membrane potential and could activate Pink1-Parkin sequentially, and then might result in ITF2 ubiquitination. In our manuscript, apart from this, we observed a huge ITF2 protein reduction in carcinoma regions of the CAC tissues and also found significant ITF2 mRNA loss in UC-associated carcinoma tissues compared to non-UC tissues in the microarray dataset (Figure 7C). Especially, the patients who experienced severe CAC clinical symptoms (stages 3 and 4) presented much lower ITF2 levels compared to the early stages

(stages 0-2) (Supplementary Figure S15B). We, therefore, might speculate that ITF2 seems to be genetically lost during the dysplasia-to-carcinoma transition as sporadic CRC did^{8,9} rather than Parkin-dependent mechanisms

References

1. Denison SR, *et al.* Alterations in the common fragile site gene Parkin in ovarian and other cancers. *Oncogene* **22**, 8370-8378 (2003).
2. Pawlyk AC, *et al.* Novel monoclonal antibodies demonstrate biochemical variation of brain parkin with age. *J Biol Chem* **278**, 48120-48128 (2003).
3. Gladkova, C., Maslen, S. L., Skehel, J. M. & Komander, D. Mechanism of parkin activation by PINK1. *Nature* **559**, 410-414, doi:10.1038/s41586-018-0224-x (2018).
4. Vyas, S., Zaganjor, E. & Haigis, M. C. Mitochondria and Cancer. *Cell* **166**, 555-566, doi:10.1016/j.cell.2016.07.002 (2016).
5. Carew, J. S. & Huang, P. Mitochondrial defects in cancer. *Mol Cancer* **1**, 9, doi:10.1186/1476-4598-1-9 (2002).
6. Boland, M. L., Chourasia, A. H. & Macleod, K. F. Mitochondrial dysfunction in cancer. *Front Oncol* **3**, 292, doi:10.3389/fonc.2013.00292 (2013).
7. Klos, P. & Dabravolski, S. A. The Role of Mitochondria Dysfunction in Inflammatory Bowel Diseases and Colorectal Cancer. *Int J Mol Sci* **22**, doi:10.3390/ijms222111673 (2021).
8. Shin, H.W., Choi, H., So, D., Kim, Y.I., Cho, K., Chung, H.J., Lee, K.H., Chun, Y.S., Cho, C.H., Kang, G.H., Kim, W.H. & Park, J.W. ITF2 prevents activation of the beta-catenin-TCF4 complex in colon cancer cells and levels decrease with tumor progression. *Gastroenterology* **147**, 430-442 e438 (2014)

9. Herbst, A., Bommer, G.T., Kriegl, L., Jung, A., Behrens, A., Csanadi, E., Gerhard, M., Bolz, C., Riesenberg, R., Zimmermann, W., Dietmaier, W., Wolf, I., Brabletz, T., Goke, B. & Kolligs, F.T. ITF-2 is disrupted via allelic loss of chromosome 18q21, and ITF-2B expression is lost at the adenoma-carcinoma transition. *Gastroenterology* 137, 639-648, 648 e631-639 (2009).

Question 5. In Fig. 5C, the authors estimate the mechanism by which ITF2 suppresses the nuclear translocation of p65 by binding to p65, but this is an unstimulated state and cannot be evaluated. It is necessary to analyze the dynamics of p65 associated with inflammatory cytokine stimulation as mentioned above.

Answer: We thank the reviewer for pointing out this critical issue. As the reviewer mentioned in question 2, we exploited ITF2-overexpressed stable cell lines (HCT116) and performed a nuclear fraction experiment with or without TNF, and used ET (empty vector) for control. TNF incubation induced p65 translocation into the nucleus whereas the levels of translocated p65 were decreased when the cells are transfected with ITF2 in the TNF-treated condition, suggesting that ITF2-p65 complex in the cytosol attenuates p65 activation under inflammatory conditions.

We included this data in Figure 5D and modified the manuscript as follows;

Manuscript Page 11, Line 239,

..Fractionation of cytoplasmic and nuclear components

HCT 116 stable cell lines established with small hairpin RNA targeting ITF2²⁰ were used and treated with or without TNF (Peprotech, Rocky Hill, CT, USA). The nuclear fraction experiment was performed as previously described.³²

...We also conducted a similar experiment under TNF-treated conditions to mimic inflammatory conditions. We observed that plenty of p65 was left in the cytosol even after TNF treatment, implying that p65 has a chance for physiological binding with ITF2 in the cytosol under inflammatory conditions. Most importantly, the amount of translocated p65 to the nucleus by the TNF challenge was downregulated by the ITF2 overexpression (Figure 5D)...

In Figure 5D,

D

Question 6. In Fig. 6, shows p65 and parkin expression with IB and IHC. Differences in NF- κ B target genes expression and cell death induction in tumor and non-tumor areas also need to be analyzed.

Answer: Following the reviewer's suggestion, we included the immunoblot of the p65 and Parkin in Figure 6B and added Parkin IHC results in Supplementary Figure S12.

For the second question, we decided to use TNF and IL-6 as NF- κ B target genes and stained cleaved caspase-3 to measure the programmed cell death population in both tumor and non-

tumor areas. As expected, TNF and IL-6 expressions were significantly increased in the tissues from ITF2^{ΔIE} compared to the littermate tissues. These phenomena were observed in both tumor and non-tumor regions (Supplementary Figures S12A, B). The expressions of Parkin and Cleaved caspase-3 seem to be upregulated in tumor areas compared to the non-tumor, but the expression levels were comparable between WT and ITF2^{ΔIE}.

We added these data in Figure 6B and Supplementary Figure S12, and modified the manuscript as follows;

Manuscript Page 23, Line 533,

...Consistently, the NF-κB target genes such as IL-6 and TNF were highly increased in both non-tumor and tumor areas of the ITF2^{ΔIE} mice in contrast with the littermates. However, Parkin and cleaved caspase-3 (one of the apoptosis markers) expression levels were comparable across the samples (Supplementary Figures S12A, B)...

In Supplementary Figure S12

In Figure 6B

Minor comments

Question 1. Please specify the TNF- α concentrations used for Fig. 1B and 2B.

Answer: We apologize for providing insufficient information. We stimulated the cells with TNF (20ng/ml) for both experiments. We specified the TNF concentration as follows;

In Figure 1,

... colonic organoids treated with TNF (20ng/ml) in.....

In Figure 2,

...treated with or without TNF (20ng/ml)....

Question 2. Please specify the cells used in Fig 4B.

Answer: Thank you for the kind comment. We specified the cell information in the figure legend as follows;

In Figure 4,

.... HEK293T cells were transfected.....

REVIEWER COMMENTS

Reviewer #1 (Remarks to the Author):

The authors have addressed my concerns, and I am satisfied with the revised manuscript.

On a minor point, on page 24, "disease severity" should be replaced by "disease stage". I would also encourage the authors to note in the discussion that further work will be required to clinically validate some of the molecular details of their findings.

Reviewer #2 (Remarks to the Author):

The authors have satisfied my prior concerns with either clarification, additional methodological descriptions, or data.

Reviewer #3 (Remarks to the Author):

In the revised manuscript, the authors have responded to most of the points raised previously and made significant improvements. I believe there is greater certainty about the interaction of ITF2 with p65, parkin-dependent ITF2 degradation, and the contribution of this cellular mechanism in CAC progression. However, there are still some points that have not been clearly resolved.

Major comments:

1. On lines 393-399; Although the authors have revised and newly added the results in Fig. 1C, D, and S3, these results do not clearly show that NF- κ B inhibitors suppress the expression of ITF2 and p65. I would like the authors to quantify how much ITF2 is increased by TNF- α stimulation compared to controls, and how much ITF2 and p65 are suppressed by NF- κ B inhibitors.
2. I pointed out last time, but it is puzzling that the total amount of I κ B α was not detected in the right panel of Fig. 2E, but phosphorylated I κ B α was strongly detected and remained without undergoing proteasomal degradation.
3. The authors show in Fig. 2G that in the presence of MG132, ITF2 is protected from proteasomal degradation and its intracellular amount is increased in analyzed cells, including WiDr cells. Furthermore, the authors showed Parkin expression in new Fig. S2, but almost no Parkin expression was observed in WiDr cells. Does this suggest that E3s other than Parkin lead ITF2 to proteasomal degradation in WiDr cells? Does endogenous ITF2 in HeLa cells stabilize in the presence of MG132? It is necessary to investigate whether only Parkin contributes to ITF2 ubiquitination as an E3.
4. In Fig. 3F right panels: Flag-p65 was expressed in lane 2 in input, although it is not to be expressed in the lane profile. Moreover, it is puzzling that the binding to HA- β -CTNNB1 was enhanced, although Myc-ITF2 was hardly precipitated in lane 4.
5. A binding simulation with AlphaFold2 in Fig. 4E should show global binding prediction of ITF2 and Parkin, and highlight the region of focus in Fig. 4E.
5. Using K48-linked or K63-linked ubiquitin chain-specific antibodies, please clarify what kind of polyubiquitin chains were attached to ITF2 by Parkin.
6. In new Fig. S10 and lines 494-496, the authors showed that cytosolic binding of p65 and ITF2 is enhanced by TNF- α stimulation. Does ITF2 bind phosphorylated p65, a hallmark of NF- κ B activation?

Minor comments:

1. On line 107; IL-1B should be IL-1 β .
2. On line 406; "Supplementary Figures S3A-D" should be S4A-D.
3. In Supplementary Figure S7A; Although it represents as Myc-ITF2B, it seems to be p65.
4. In Fig. 3C, it is not clearly stated which region the Δ NT contains.

Responses to the reviewers' comments regarding our manuscript entitled "Protein Stabilization of ITF2 by NF- κ B Prevents Colitis-associated Cancer Development"

(Manuscript ID: NCOMMS-21-47878A)

Statement of Revision to Reviewer #1

We would like to begin by expressing our appreciation for the reviewer's thoughtful and considerate suggestions and comments. We hope our responses will be deemed satisfactory. Thank you for the reviewer's critical reading and suggestions for strengthening our conclusions.

Minor comments

Question 1. on page 24, "disease severity" should be replaced by "disease stage". I would also encourage the authors to note in the discussion that further work will be required to clinically validate some of the molecular details of their findings.

Answer: We appreciate the reviewer's valuable comments. Following the reviewer's suggestion, we changed the wording "disease severity" to "disease stage" on page 24.

We revised the manuscript as follows;

Manuscript Page 24, Line 560-564,

..... **disease stage** in patients with UC and CAC. After obtaining additional 16 UC specimens, we stained ITF2 and then compared the **disease stage** based on ITF2 expression levels. Importantly, the patients in the late stage (stages 3 and 4) presented significant loss of ITF2 (Supplementary Figures S15A, B), suggesting that ITF2 is highly correlated with CAC **disease stage**.....

In addition, we described those points in the discussion part as follows;

Manuscript Page 28, Line 659,

... Our study has identified a potential tumor suppressor gene, ITF2, which may play a role in inhibiting inflammation-mediated cancer progression. Based on the previous microarray dataset (Figure 7C), UC-associated neoplastic lesions, UC-associated carcinoma tissues, and sporadic CRC tissues displayed rare ITF2 mRNA expressions, suggesting that the levels of ITF2 have already disappeared in the mRNA levels. We acknowledge that there are limitations to our findings, such as the lack of validation on the specific factors or genetic mutations that could be involved in downregulating ITF2 expression during the dysplasia-to-carcinoma transition. Therefore, further studies are needed to examine the molecular details of our findings on a larger scale, including genetic mutations in ITF2, SNPs in specified genes, and post-translational modification of ITF2...

Statement of Revision to Reviewer #2

We authors thank the reviewer for the comments, which contributed to the improvement of this manuscript.

Statement of Revision to Reviewer #3

We thank the reviewer for raising these issues and hope that our response will be satisfactory.

Major comments

Question 1. On lines 393-399; Although the authors have revised and newly added the results in Fig. 1C, D, and S3, these results do not clearly show that NF- κ B inhibitors suppress the expression of ITF2 and p65. I would like the authors to quantify how much ITF2 is increased by TNF- α stimulation compared to controls, and how much ITF2 and p65 are suppressed by NF- κ B inhibitors.

Answer: We appreciate the reviewer's comment. To clarify the results, we included an additional cell line (HCT15), which expresses ITF2 at low levels. We quantified the levels of specified proteins, such as ITF2 and p65, in each condition using Image J software. The band intensities were normalized to the corresponding tubulin levels, which served as a loading control. Although the degree of increase in ITF2 and p65 levels by TNF- α and the extent of suppression by NF- κ B inhibitors differed somewhat in each cell line (Colo320DM, SW480, WiDr, HCT15), the quantification results suggest that ITF2 levels are upregulated approximately two-fold by TNF- α challenge, but decreased around half by p65 inhibitors. P65 levels themselves did not show significant downregulation but exhibited a decreased pattern upon p65 inhibitor treatment. As the TNF- α treatment did not significantly induce p65 downregulation, we have revised the wording to be more tone-downed manner as follows;

Manuscript Page 17, Line 394,

... The SW480, WiDr, and HCT15 cell lines including Colo320DM which can express ITF2 showed a similar result (Figure 1D and Supplementary Figures S3A, B). Although NF- κ B inhibitors did not result in a noteworthy decrease in p65, they exhibited a reduction in a pattern following p65 inhibition, as shown in Figures 1C, D, and Supplementary Figure S3.... Thus, the question arose of whether p65 itself induces ITF2 expression.

We added quantification results in Figure 1D and Supplementary Figures S3A, B and then revised the manuscript as follows;

In Figure 1D,

In Figure S3,

Figure Legend 1D (Manuscript Page 30, Line 677)

..... immunoblots. Each data are expressed as normalized band intensity adjusted to α -tubulin, which serves as a loading control. The protein intensities were quantified by ImageJ software....

Figure Legend S3 (Supplementary manuscript Page 9, Line 160)

....(A) SW480, WiDr and HCT15 cells were pre-incubated with indicated inhibitors for 1 h followed by TNF stimulation for 8 h. The indicated proteins were traced and presented as

immunoblots. (B) Each data are expressed as normalized band intensity adjusted to α -tubulin, which serves as a loading control. In all immunoblot analyses, protein intensities were quantified by ImageJ software. Statistical significance for normalized band intensity was determined by Student's *t*-test. **P* < 0.05; ***P* < 0.01. Data are shown as means \pm s.e.m....

Question 2. I pointed out last time, but it is puzzling that the total amount of I κ B α was not detected in the right panel of Fig. 2E, but phosphorylated I κ B α was strongly detected and remained without undergoing proteasomal degradation.

Answer: We are sorry about this confusion. First, we would like to provide the reviewer with a clear description of our experimental design as presented in the manuscript.

The samples displayed in Figure 2E were already pre-incubated with TNF- α for 8 h prior to the time-dependent CHX treatment. This pre-treatment led to the upregulation of both ITF2 and p-I κ B α levels as shown in the first lane of the right panel compared to the first lane of the left panel. As the reviewer knows, the levels of I κ B α (total form) decreased by the TNF- α challenge as they phosphorylated by the TNF- α stimulation. That is why the total amount of I κ B α seems to be invisible in the TNF- α -stimulated group. However, to address any potential confusion, we have changed the immunoblot band of I κ B α to a more visible one (long exposure version).

We have updated the relevant section in the manuscript to reflect this finding as follows;

In Figure 2E,

E

Question 3. The authors show in Fig. 2G that in the presence of MG132, ITF2 is protected from proteasomal degradation and its intracellular amount is increased in analyzed cells, including WiDr cells. Furthermore, the authors showed Parkin expression in new Fig. S2, but almost no Parkin expression was observed in WiDr cells. Does this suggest that E3s other than Parkin lead ITF2 to proteasomal degradation in WiDr cells? Does endogenous ITF2 in HeLa cells stabilize in the presence of MG132? It is necessary to investigate whether only Parkin contributes to ITF2 ubiquitination as an E3.

Answer: Thank you for the meticulous comments.

In the first experiment, we investigated whether MG132 could stabilize ITF2 levels in HeLa cells, which have a low amount of Parkin. Our findings showed no significant differences between the samples treated with vehicle and MG132. This result appears to be reasonable, given the low levels of Parkin in HELA cells.

In relation to the second question, we agree with the reviewer's suggestion that WiDr cells have relatively lower levels of Parkin compared to other ITF2-expressing cell lines, such as Colo320DM, SW480, and CaCo2, as shown in Figure S2A. However, upon closer inspection of the long exposure version of the Parkin immunoblot band, we observed enough amount of Parkin in WiDr cells, despite their lower levels compared to the other cell lines.

We surmise that the quantity of E3s present does not necessarily correspond to the levels of substrate degradation. Instead, if there are enough E3s available, they can still play a role in the cells. It is still possible that other E3s, rather than Parkin, could lead to ITF2's proteasomal degradation in WiDr cells, as suggested by the reviewer. Nonetheless, based on the results of our Parkin knockdown experiment in WiDr cells, we can at least propose that Parkin contributes to ITF2 ubiquitination in these cells. We hope that our responses are satisfactory.

Question 4. In Fig. 3F right panels: Flag-p65 was expressed in lane 2 in input, although it is not to be expressed in the lane profile. Moreover, it is puzzling that the binding to HA- β -CTNNB1 was enhanced, although Myc-ITF2 was hardly precipitated in lane 4.

Answer: We apologize for the error and have revised the lane profiles as per the reviewer's comment. The lane profiles of the Flag-p65 on the right should correspond to the lane profiles of the HA-CTNNB1 on the left. Conversely, the lane profiles of the HA-CTNNB1 on the right should match the lane profiles of the Flag-p65 on the left since this experiment is a competitive binding assay involving ITF2. We have revised the lane profiles based on the reviewer's feedback.

Regarding the second question, our study demonstrates that ITF2 binds with p65 and is stabilized at the protein level. Given that p65 and β -CTN competitively bind to ITF2, we hypothesize that

increased binding of β -CTN to ITF2 may decrease the probability of p65 binding to ITF2, leading to reduced levels of stabilized ITF2 and potentially affecting the levels of precipitated ITF2. We hope this explanation addresses your inquiry.

In Figure 3F,

Question 5. A binding simulation with AlphaFold2 in Fig. 4E should show global binding prediction of ITF2 and Parkin, and highlight the region of focus in Fig. 4E.

Answer: In response to the reviewer's suggestion, we have made an addition to Figure 4E by including a global binding prediction result generated by AlphaFold2 on the left and highlighting the specific region of interest (right). The updated figure now includes;

In Figure 4E,

E

Question 6. Using K48-linked or K63-linked ubiquitin chain-specific antibodies, please clarify what kind of polyubiquitin chains were attached to ITF2 by Parkin.

Answer: We would like to express our gratitude to the reviewer for bringing up an important matter. As per the reviewer's recommendation, we conducted an investigation to identify which lysine (K) residues in the ubiquitin molecule could be involved in Parkin-mediated ITF2 ubiquitination. We utilized two types of antibodies specific to K48 or K63 and found that the K48 ubiquitin chain was utilized for the Parkin-mediated ITF2 ubiquitination.

We have updated the relevant section in the manuscript to reflect this finding as follows;

Manuscript Page 21, Line 483

... More specifically, the K48-linked ubiquitin chain is engaged in the Parkin-mediated ITF2 ubiquitination (Figure 4F)....

In Figure 4F,

Question 7. In new Fig. S10 and lines 494-496, the authors showed that cytosolic binding of p65 and ITF2 is enhanced by TNF- α stimulation. Does ITF2 bind phosphorylated p65, a hallmark of NF- κ B activation?

Answer: In order to confirm the binding between ITF2 and p-p65, we conducted an immunoblot experiment using the same samples as those in Figure S10D. As anticipated, the levels of immunoprecipitated p-p65 were found to be elevated in the samples treated with TNF- α . We have included this figure and made the necessary modifications to the manuscript as follows;

.... In addition, the levels of immunoprecipitated p-p65, a hallmark of NF- κ B activation, by the ITF2 are augmented after TNF treatment (Supplementary Figure S10E)...

In supplementary Figure S10E

E

Minor comments

Question 1. On line 107; IL-1B should be IL-1 β .

Answer: We corrected the reference to "IL-1B" as "IL-1 β " as suggested by the reviewer.

Question 2. On line 406; "Supplementary Figures S3A-D" should be S4A-D.

Answer: The wording "S3A-D" has been changed to "S4A-D" as requested.

Question 3. In Supplementary Figure S7A; Although it represents as Myc-ITF2B, it seems to be p65.

Answer: Thank you for looking through all figures very carefully. However, if we take a look at the expression patterns of ITF2B, ITF2B (NT), and ITF2B(Δ NT) in Figure 3C, their expression patterns are exactly corresponding to the bands shown in Supplementary Figure S7A. So, the authors think that the bands shown in S7A represent ITF2B.

Question 4. In Fig. 3C, it is not clearly stated which region the Δ NT contains.

Answer: We apologize for the confusion. The ITF2 mutant (Δ NT) was generated by deleting the regions between (1-250) of the whole ITF2 domains, as denoted at the top of Figure 3C.

REVIEWERS' COMMENTS

Reviewer #3 (Remarks to the Author):

The authors have performed appropriate modifications and additional experiments, and there are no concerns.